# Axial induction controller field test at Sedini wind farm

Ervin Bossanyi[1], Renzo Ruisi[1]

[1]DNV GL, One Linear Park, Avon Street, Bristol, BS2 0PS, UK

*Correspondence to*: Ervin Bossanyi (ervin.bossanyi@dnvgl.com)

**Abstract.** This paper describes the design and testing of an axial induction controller implemented on a row of nine turbines on the Sedini Wind Farm in Sardinia, Italy. This work was performed as part of the EU Horizon 2020 research project CL-Windcon. An engineering wake model, selected for its good fit to historical SCADA data from the site, was used in the LongSim code to optimise turbine power reduction setpoints for a large matrix of steady-state wind conditions. The setpoints were incorporated into a dynamic control algorithm capable of running on site using available wind condition estimates from the turbines. The complete algorithm was tested in dynamic time-domain simulations using LongSim, using a time-varying wind field generated from historical met mast data from the site. The control algorithm was implemented on site, with the wind farm controller toggled on and off at 35-minute intervals to allow the performance with and without the controller to be compared in comparable wind conditions. Data was collected between July 2019 and early February 2020. The results have been analysed and indicate a positive increase in energy production resulting from the induction control, in line with LongSim model predictions, although a larger volume of valid data would be necessary to provide statistically robust conclusions. The measurements also provide a validation of the LongSim model, proving its value for both steady state setpoint optimisation and time-domain simulation of wind farm performance.

## 1 Introduction

Wake interactions are well known to reduce wind farm power output and increase turbine loads. Recent years have seen much interest in wind farm control concepts aimed at reducing these wake effects. The control objective is to increase overall wind farm power production while maintaining or reducing turbine fatigue loads, by manipulating the individual turbine controllers to minimise wake interaction effects, using either axial induction control or wake steering. Both control concepts involve deliberately reducing the power output of some individual turbines in order to achieve a net increase in total production from the farm. In the case of axial induction control, turbine power reduction is achieved by increasing the pitch angle and/or reducing rotor speed in order to reduce rotor thrust, thus weakening the wake. In wake steering control, the turbine is deliberately yawed out of the wind direction at angles typically up to 30 degrees, as this has the effect of changing the downstream path of the wake, which can thus be steered away from downstream turbines.

Axial induction control has been investigated using large eddy simulation modelling, often without showing any positive gains in power production – see for example Gebraad (2014). It has since been tested in a boundary layer wind tunnel by

Campagnolo et al. (2016a, 2016b) as well as in an operational wind farm by Van Der Hoek et al. (2019). During the wind tunnel tests no net gains were obtained, but an increase in power production has been reported during the field tests compared to standard operation. As reported in van Wingerden et al. (2020), a survey among technical experts of wind farm control highlighted how the need for increased confidence in modelling the effects of wind farm control via more validation campaigns was seen as the top priority.

As part of the EU Horizon 2020 research project CL-Windcon (www.clwindcon.eu) two field test experiments were designed and carried out at the Sedini Wind Farm in Sardinia, Italy, in order to test the two main concepts for active wake control in wind farms (axial induction and wake steering). This paper specifically reports on the axial induction control tests. Further details of all the tests can be found in Kern et al (2019). The results of the wake steering field tests during the same measurement campaign are reported in Doekemeijer et al. (2020).

Section 2 presents an overview of the Sedini wind farm site and the planning of the induction control experiment. The initial controller design process is described in Section 3. In Section 4, the use of time-domain simulation modelling to test and refine the controller is described, while the field tests themselves are described in Section 5, and results are presented.

## 2 The Sedini wind farm site

Details of the Sedini onshore wind farm, planned instrumentation and test campaigns are provided in Schuler et al (2017).
The farm consists of 43 GE 1.5 turbines laid out as in Figure 1. Most of the turbines are of type GE 1.5s (1.5 MW, 70.5m rotor diameter, 65 m hub height), but the seven turbines shown in red are the larger GE 1.5sle (1.5 MW, 77 m rotor diameter, 80 m hub height). The diagonal row of turbines 13 and 31 – 38 is involved in the experiment described here, and since only wind directions blowing along this row from a roughly south-westerly direction are relevant to the experiment, only these nine turbines were modelled in the controller design phase. Terrain complexity has been ignored – the site is not completely
flat, but the topography indicates that with south-westerly wind directions, the effect of the terrain on the wind flow at these nine turbines is likely to be relatively small. The wind rose in Figure 1 shows a preponderance of westerly wind during the field test period, with relatively little from the south-west.

The original intention was to carry out both induction and wake steering field tests using this row of turbines. Preliminary design work for both sets of tests is documented in Knudsen et al (2019).  However, because of instrumentation issues, only
the induction control tests were actually carried out, and this paper describes the final controller design and simulation testing, and presents results from the field tests which began in July 2019. A separate test of wake steering control was carried out by yawing turbines 26 and E5, as described in Kern et al (2019).

Since no loads instrumentation was available on the turbines used for the induction control experiment, the induction control is aimed only at increasing the total power production from this row of turbines. The power output of turbines 31 – 37 can be

modified, and the power output of all nine turbines is monitored. Turbine 38 is used as a reference turbine and wind sensor, and it remains in baseline operation. Some additional gain might be expected if turbine 38 were also controlled, but this has been sacrificed to ensure that the accuracy of the wind estimation is not affected by any control action. Turbine 13 is not controlled as there are no turbines in its wake, but clearly its power output will be affected.

During the field tests, the wake control is switched on and off at regular intervals (determined as in Section 3.5) so that the
performance with and without control can be compared in similar wind conditions.

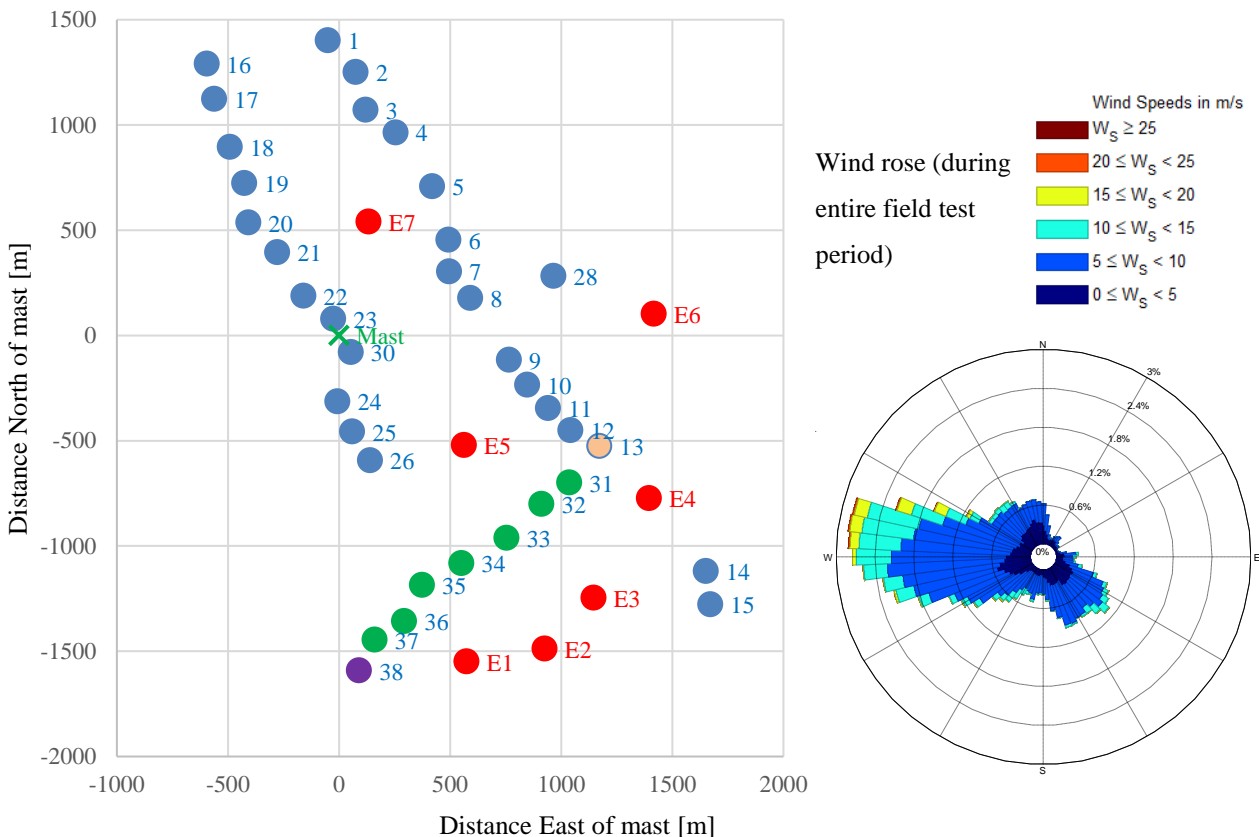

**Figure 1: Site layout. Induction control field test involves turbines 13 and 31-38, with winds from the south-west. The set-point optimisation maximises the total power from these nine turbines. Controlled turbines are in dark green. Turbine 38 is used as the reference from which wind conditions are calculated. Turbine 13 is affected but not controlled, as its wake does not affect other turbines.**

### 3 Controller design

The design work was carried out using the LongSim code. This has been developed by DNV GL, and more details can be found in Bossanyi et al (2018). It is used for the initial steady-state setpoint optimisation, described in Section 3.2, and also

for the dynamic time-domain simulation testing described in Section 4. For illustration, a typical wind speed contour plot at one point in time during a dynamic simulation is shown in Figure 2.

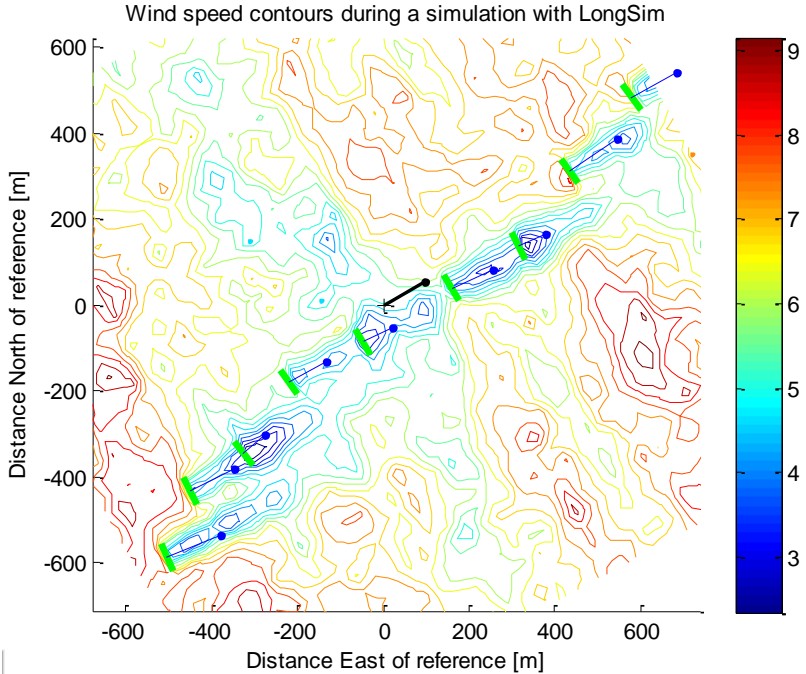

**Figure 2: Typical wind speed contour plot from a LongSim dynamic simulation. Rotor planes of the nine turbines #13 and #31 to #38 of the Sedini Wind Farm are represented as thick green lines, with blue lines pointing along the local wind direction with length proportional to the local wind speed magnitude.**

## 3.1 Wake modelling

To allow rapid calculations and design iterations, LongSim does not use high-fidelity flow modelling, but makes use of fast engineering wake models embedded in an ambient flow field. A choice of different engineering models is available, and for the preliminary design reported in Knudsen et al (2019), several different wake models were used to investigate the sensitivity of the wake control performance to the wake model details, and it was clear that the wake model can make a big difference to the results. In this section, historical SCADA data from Sedini is used to help in the selection of a single wake model to be used in the final controller design.

SCADA data recorded from 01/05/2018 to 05/03/2019 was processed to extract the 10-minute average power output for each of the nine turbines, and the ratio of power at each turbine #13 and #31-#37 to the power of the reference turbine #38 was plotted as a function of wind direction. The power ratio for any turbine showed a clear dip for any wind directions where the turbine was affected by a wake. For each turbine, as shown in Figure 5, the power ratios were binned in 5° bins and the mean and median ratio in each bin was calculated. The median was found to be more useful than the mean, as it avoids big spikes

caused by outliers in the data (see first plot of Figure 5 for example). Each candidate wake model was used to calculate a predicted power ratio for the direction corresponding to the middle of each bin (blue lines in Figure 5), and the RMS errors between the median and the predicted values were summed over the direction bins and then over all the turbines #13 and #31-#37 to give a measure of the goodness of fit for this wake model. The RMS errors for the different candidate wake models are shown Figure 4. All the wake models are implemented within the LongSim code, which was used to generate the results presented here.

The candidate wake models included the EPFL model of Bastankhah and Porté-Agel (2016) and several variants of the model of Ainslie (1988). The EPFL model includes a number of parameters which many researchers have subsequently used as tuning parameters, adjusted to fit particular datasets, as has also been done within the CL-Windcon project (for example, the model was calibrated against wind tunnel measurements in Raach et al, 2018). Here, only the original parameters specified in Bastankhah and Porté-Agel (2016) were used and no attempt was made to tune them. It is likely in any case that different parameters would work best for different conditions of, for example, atmospheric stability, so it is more useful if a general model can be found which does not rely on such tuning. The Ainslie model is treated as such a general model, in that the parameters defining the wake deficit profile and its downstream expansion are considered fixed, but a number of variations are still possible. In particular, the following variations of the basic Ainslie model were investigated here:

a) Choice of wake-added turbulence model: either the Crespo-Hernández model (CH) as assumed in the EPFL model, or the Quarton-Ainslie model (QA) as used, for example, by WindFarmer (DNVGL, 2014),

b) Choice of wake superposition models: the dominant wake model (DW) in combination with 'large wind farm' corrections as in WindFarmer (DNVGL, 2014), or the sum-of-absolute-deficits model (SD) as in Ruisi and Bossanyi (2019),

c) Accounting explicitly for hub height in the modelling of the eddy viscosity parameter (the original model only uses the rotor diameter),

d) More precise calculation of centreline deficit, using momentum conservation to avoid having to integrate over a radially-discretised flow (Anderson, 2019),

e) Wake smearing to account for the effect of wake meandering over the averaging time as in Bossanyi et al (2018),

f) Modification of the eddy viscosity term to account for atmospheric stability as in Ruisi and Bossanyi (2019).

In respect of the last point, met mast data from the site was analysed to identify diurnal variations in the wind conditions, driven by predominant unstable and stable conditions during the daytime and night-time hours respectively, and estimate bulk Richardson numbers and correspondent Obukhov lengths (these two parameters are defined and discussed in Ruisi and Bossanyi (2019)) to classify atmospheric stability conditions occurring at the site. A summary of the atmospheric stability conditions by time of day at the site is shown in Figure 3. Given this information, the recently-developed stability-dependent

eddy-viscosity model of Ruisi and Bossanyi (2019) was used, allowing the effect of atmospheric stability to be directly accounted for. The SCADA data was split into three different classes based on the time of day: daytime (hours 7 – 17), night-time (hours 18 – 06), and overall. In the daytime the atmosphere is generally unstable, with an average historical Obukhov length of -255m (the negative value signifying unstable conditions), while the night-time period is generally stable, with an average historical Obukhov length of 237m. The overall average Obukhov length was 850m.

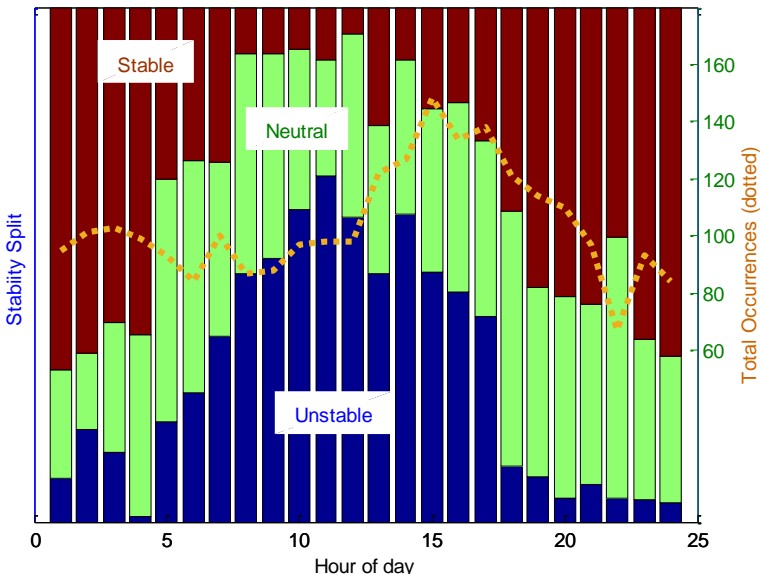

**Figure 3: Diurnal distribution of atmospheric stability conditions, classified into three categories based on the Bulk Richardson number estimated from the site mast at the Sedini Wind Farm site.**

The Ainslie model variations as detailed in a) to f) above correspond to the labels in Figure 4 as follows:

| | a) | b) | c) | d) | e) | f) | |
|---|---|---|---|---|---|---|---|
| AinslieStandard | QA | DW | | | | | |
| AinslieMOL | CH | SD | √ | | | √ | |
| AinslieMOL_QA | QA | SD | √ | | | √ | |
| (AinslieSP4 | QA | * | | | | | *An experimental superposition model, since abandoned) |
| AinslieSumOfDefs | QA | SD | | | | | |
| Ainslie_H_SoD | QA | SD | √ | | | | |
| Ainslie_H_SoD_Exact | QA | SD | √ | √ | | | |
| AinslieMOL_QA_Exact | QA | SD | √ | √ | | √ | |
| AinslieMOL_QA_WS_Exact | QA | SD | √ | √ | √ | √ | |

The comparison of wake models in terms of overall RMS error is shown in Figure 4. The model selected for the final design is the one labelled "AinslieMOL_QA_Exact", which has the lowest overall error for both daytime and night-time periods, and nearly the lowest overall. This is the stability-dependent variant of the Ainslie model (Ruisi and Bossanyi, 2019), together with Quarton-Ainslie added turbulence, sum-of-absolute-deficits superposition, explicit hub height, and the more

precise centreline deficit calculation (these options are described above). Other variants of the Ainslie model are available, differing from one another in subtle points of detail. Using the selected model with the Obukhov length for averaged-neutral conditions, the fit against the SCADA data is shown in Figure 5 for each of the turbines.

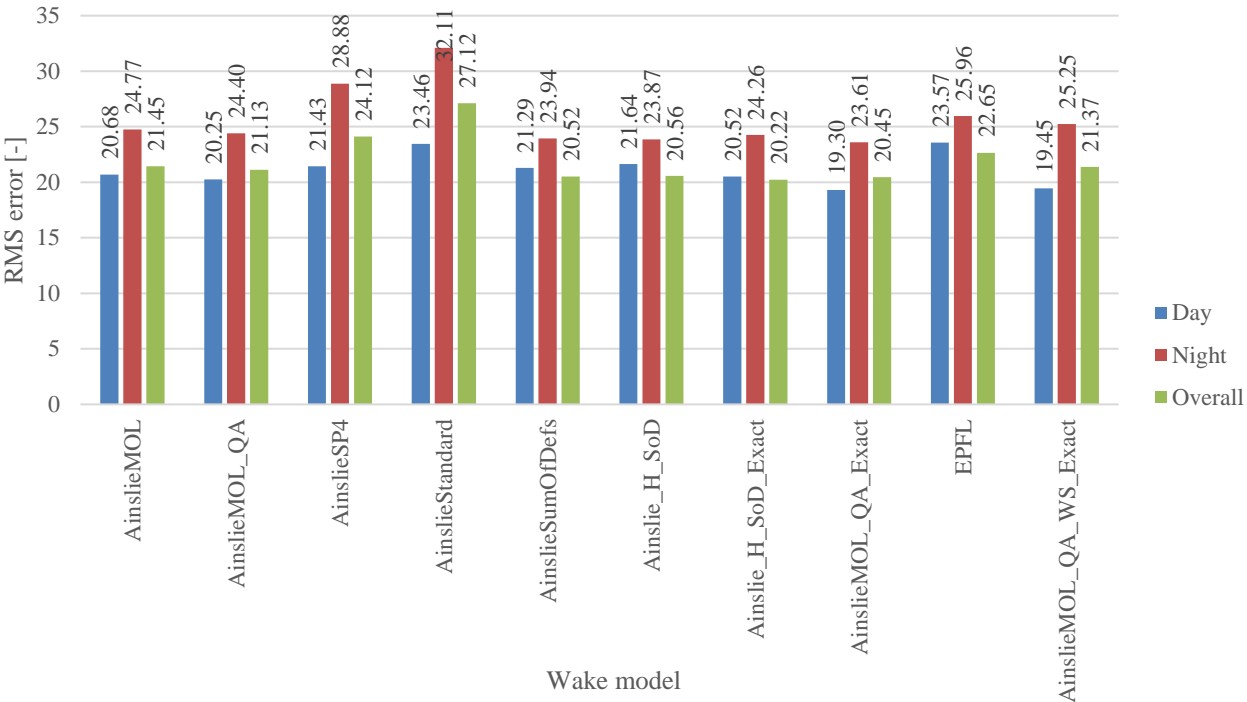

**Figure 4: Overall comparison of different wake models.**

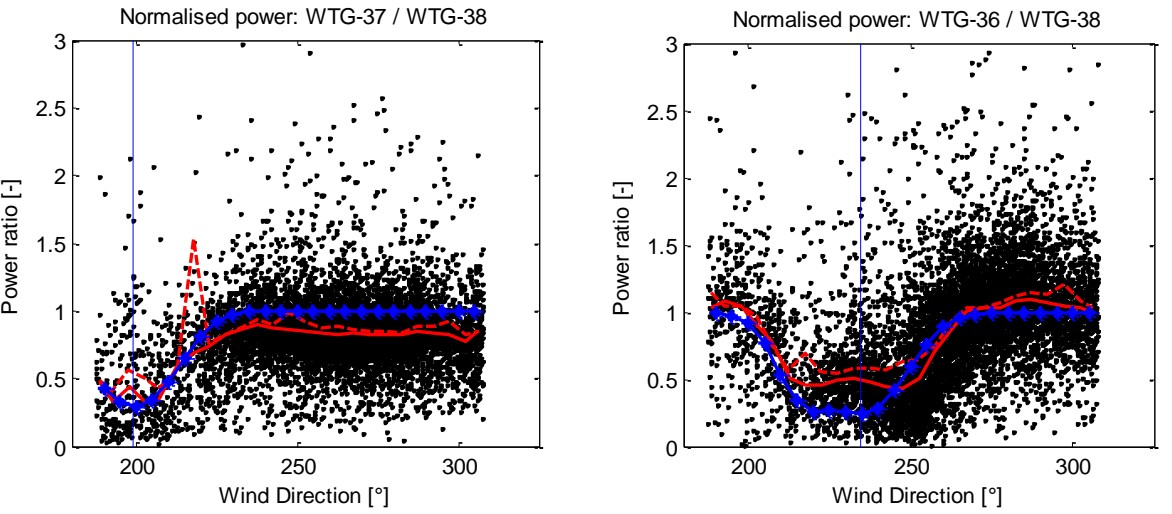

**Figure 5 (continued on next page …)**

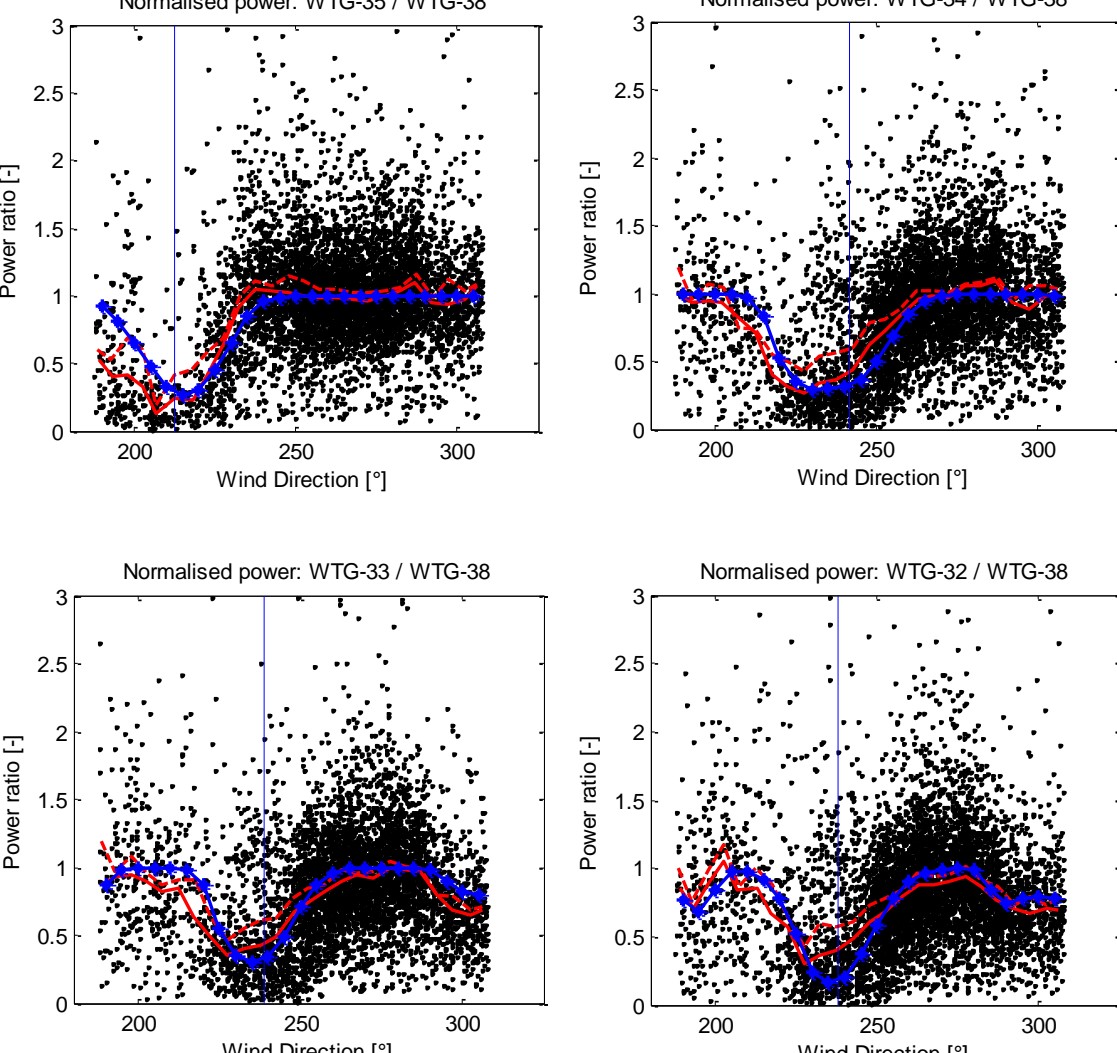

**Figure 5 (… continued …)**

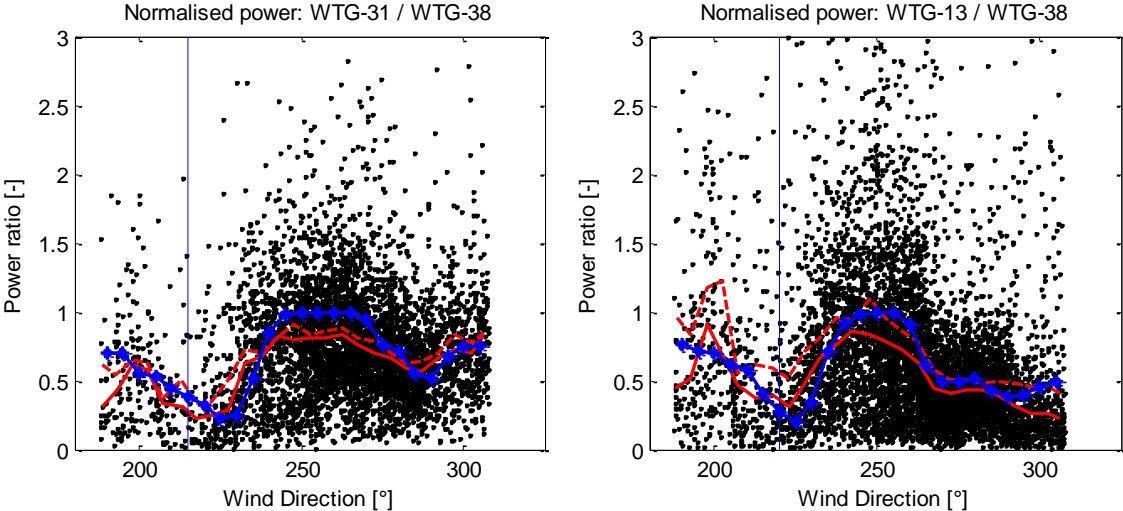

**Figure 5 (concluded): Selected model (blue) compared to SCADA data (black, with bin means shown dashed red and medians solid red). The vertical blue line shows the direction of the turbine just upstream.**

### 3.2 Steady-state setpoint optimisation

Since the selected wake model includes a dependence on atmospheric stability, it would be possible to calculate optimal setpoints for different Obukhov lengths, and to use a measurement of the Obukhov length to modify the setpoints in real
time, as will already be done for wind speed, direction and turbulence intensity. However, for the purposes of the Sedini experiment this would not be possible to arrange, and so the setpoints were calculated using the average Obukhov length of 850m derived from the historical data, representing near-neutral conditions. A further improvement to the results would have been likely if it had been possible to use measured stability as a lookup table input.

Using this wake model, the steady-state optimiser in LongSim was then used to generate tables of optimised power setpoints
for each controlled turbine, i.e. #31 to #37. The merit function for optimisation was the total power from all 9 turbines, i.e. also including #38 and #13. Setpoints were calculated for wind speeds from 6 to 15 m/s in 1m/s steps, directions from 200 to 270 degrees in 2-degree steps, and turbulence intensities of 9, 13 and 17%. The speed and direction ranges in the tables were extended to 3-18 m/s and 180-270 degrees by padding with null setpoints (i.e. no power reduction). The final look-up table (LUT) consists of setpoints as a function of wind speed, direction, turbulence intensity and turbine number.

The effect on turbine loads is also important, and in general the merit function could include terms related to loads. However, this has not been done since there was no possibility within the project to measure loads on these turbines at Sedini. In general, most loads are generally expected to decrease anyway with axial induction control, both on controlled and on downstream turbines, although the pitch actuator duty cycle would increase because of the below-rated pitch action.

The following sections describe how the resulting LUT was converted into a practically realisable control algorithm.

## 3.3 Measurement of the wind condition

For practical application, the controller needs to have an estimate of wind speed, wind direction and turbulence intensity at each time step so that it can obtain the appropriate setpoints from the LUT. Since the setpoints are optimised on the assumption that the (undisturbed) wind condition is the same throughout the wind farm, wind condition estimates should be representative of the whole farm. In general, a met mast could be used if one is available, but more than one mast would be needed to cover different wind directions, so it would usually be better to use estimates from the turbine controllers. Each turbine controller can provide a direction estimate by filtering its nacelle position signal plus the wind vane misalignment, as long as suitably calibrated measurements are available. The turbine controller can usually provide a wind speed estimate, and if a separate turbulence intensity estimate is not available it can be obtained from the wind speed estimate standard deviation with appropriate calibration factors. The wind farm controller could then use the average or the median of the wind conditions estimates from all turbines which are currently unwaked, and use this to represent the whole farm. A low-pass filter can be applied with a variable time constant of the order of the time taken for a wind condition measured at the upstream edge of the farm to propagate to the middle of the farm. This introduces an appropriate delay as well as some smoothing.

For the specific row of turbines used at Sedini, the following approach was used. The upstream turbine, #38, is always unwaked in wind directions of interest and is used to estimate the wind speed and turbulence intensity, which is then used for the LUT as if it represents the whole row of turbines. The wind direction for the LUT is as taken as the median of the individual wind direction estimates provided by all nine turbines in the row. This assumes that wake effects do not change the local wind direction, which is more likely to be true for induction control than for wake steering cases.

The inflow wind speed is an estimate of the rotor averaged wind speed based on 1Hz operational data of turbine #38. The individual wind direction estimates are derived from the nacelle position sensor and the nacelle vane signals. Prior to starting the test, the nacelle position sensors signals had been calibrated using the preceding 3 months of SCADA data. The calibration process was designed such that the resulting wind direction estimates comply with the assumption that the time averaged wake velocity deficits propagate with the mean wind direction. An online algorithm ensures that the calibration of the nacelle position sensors is maintained over time in case irregularities occur.

The turbulence intensity is derived from the standard deviation of the estimated wind speed, with a correction factor applied which has been derived by comparing the standard deviation calculated in the same way at a turbine close to the met mast against the standard deviation actually measured at the mast.

The estimated wind speed and direction signals are 60s averages, while the turbulence intensities are instantaneous values from a running 10-minute estimation.

If turbine #38 is not running, the test continues using wind estimates from #37. If neither of those turbines is working, #36 is used. If all three turbines are not running, no wind farm control is applied. However, it should be noted that the optimal setpoints are only valid if all nine turbines are working. Cases with some turbines not working were not tested in simulation, and in the analysis of the field test results, data was discarded if not all turbines were working.

### 3.4 Accounting for wind condition uncertainty

The power reduction setpoints are optimised using steady-state calculations for specific ambient wind conditions which are assumed to apply over the whole wind farm. In the practical application, the wind condition used for the LUT to calculate setpoints at any specific time are not precisely known, partly because of uncertainties in measurements used, and partly because the wind conditions at any time are not uniform across the wind farm. The setpoint optimisation can already take account of such uncertainties by assuming probability distributions rather than fixed values for the wind speed, direction and
turbulence intensity used for each optimisation, as described in Rott et al. (2018) and Simley et al. (2020) for the case of robust active wake control optimisation. This results in lookup tables which are smoothed out by those probability distributions, but the time needed for the optimisations greatly increases. Here an alternative approach is used, in which the LUT calculated for precise wind conditions is smoothed out subsequently, with each value replaced by a weighted average of nearby values, the weightings being determined by those assumed probability distributions. This has the advantage of
faster optimisation, but also means that in principle the smoothing can be changed in real time according to the perceived uncertainties in wind conditions at the time.

For the field tests, this post-hoc smoothing was carried out using fixed assumptions about the uncertainties, namely that the wind speed and direction have Gaussian distributions with standard deviations of 1m/s and 5º respectively. Because of the smaller dependence of the setpoints on turbulence intensity, no smoothing was applied for turbulence intensity. Prior to field
testing, the smoothing assumptions were tested in simulation as described below.

### 3.5 Final control algorithm design

The final control algorithm updates the setpoints on a timestep of 60 seconds. At every timestep, the wind condition, estimated as described in Section 3.3, is used to generate a setpoint for each turbine using the setpoint LUT which has been smoothed as described in Section 3.4. The power reduction setpoints are then sent directly to the turbine controllers.

For the purposes of the field test, the controller is toggled on and off every 35 minutes. This toggle frequency was selected on the basis that the wind advection time along the row from #38 to #13 will be of the order of $2 - 5$ minutes in the wind speed range of interest, and a further 30 minutes before switching should be enough time to get a representative result, and the toggling should be frequent enough to obtain periods with similar wind conditions in both toggle states. Choosing 35 minutes also ensures that switching does not occur at exactly the same time every day, which could introduce a bias due to
interaction with diurnal changes in wind conditions. Data from the field tests was recorded at 1-minute intervals.

The final algorithm was tested in dynamic time-domain simulations as described in Section 4, before being implemented in the field. Section 5 describes the field test and presents an analysis of the results.

## 4 Simulation testing

Before the wind farm control was implemented in the field, dynamic simulations were run with LongSim to try to mimic the behaviour of the wind farm as closely as possible in realistic time-varying wind conditions, and to assess the likely performance of the wind farm control.

The simulations used a correlated stochastic wind field covering the turbines, generated by LongSim starting from historical data measured at the Sedini met mast, thus ensuring that at least the lower-frequency wind variations are appropriate for the site. The simulation results provided time histories of wind conditions, setpoints and power outputs at each of the turbines. Simulations were run with and without wind farm control, and also with the control toggling on and off every 35 minutes as would be done in the field.

### 4.1 Wind field

The technique for generating the correlated ambient wind field has been described in Bossanyi et al (2018). The 10-minute average historical met mast data was inspected, and a period selected where the wind speeds and directions were varying over a range suitable for exercising the wind farm control. This time history was assumed to apply at a point in the middle of the row of turbines, and higher-frequency synthetic turbulence was added at that point, and also at a grid of points covering all the turbines, using assumed coherence properties, so that variations across the wind farm are realistically correlated, spatially and in time. LongSim's default settings were used for the spectral and coherence properties of the wind.

The wind field was modified by wind shear appropriate for the site, modelled with a shear exponent of 0.143, and the air density was taken as 1.177 kg/m$^3$.

### 4.2 Turbine model

Although a detailed model of the turbine was not provided, LongSim has the option to model the turbine using power and thrust curves as a function of wind speed, which is sufficient for a basic evaluation. Power and thrust curves were provided covering the allowed range of power reduction setpoints. LongSim also models supervisory control, and in this case the yaw control algorithm provided by GE was implemented, to ensure a realistic response to changing wind directions. Figure 6 illustrates the resulting yaw response during a short example simulation.

The turbine was modelled with a 10-second first-order lag for implementation of the power reduction setpoint. This is an approximation to the actual behaviour; details of this were not provided, save that in lower winds the thrust reduction relies on a change in rotor speed, which might take a few seconds, but in higher winds only a change in blade pitch is needed,

which is faster. Simulation results confirmed that a lag of this order has only a very small effect on the induction control performance. The actual setpoint is a dimensional index number upon which the turbine controller acts to reduce both power and thrust, to an extent which varies with wind speed; details were not provided by the manufacturer for reasons of confidentiality, but the maximum reduction does not exceed 20% of rated power.

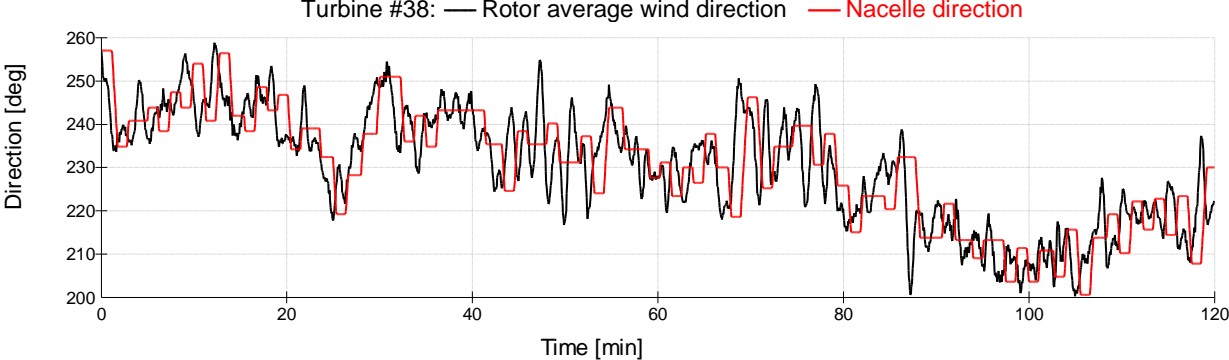

**Figure 6: Typical simulated yaw control response.**

### 4.3 Wake model

The wake model selected as described in Section 3.1 was used for the simulations. As these are dynamic simulations, assumptions also need to be made concerning the dynamic wake response. LongSim's default assumptions were used for the wake advection speed, namely that the advection speed is the average of the ambient speed and the speed integrated over the

wake. Wake meandering was driven by the low-frequency lateral and vertical components of the wind field up to a wavenumber corresponding to two turbine diameters. The resulting wakes are simply embedded into the ambient wind field, which is assumed not to be otherwise affected by the presence of the turbines.

### 4.4 Induction control algorithm

The wind farm control algorithm used the same LUT as was subsequently implemented on site. Simulations were run first

with the raw LUT, and then with the LUT corrected for wind condition uncertainties as described in Section 3.4, firstly just with 5° direction uncertainty and then with a further uncertainty of 1 m/s in wind speed.

The wind conditions for the LUT were calculated as in the site implementation, i.e. using turbine #38 for wind speed and turbulence intensity and all nine turbines for direction, but ignoring any inaccuracy in the estimations, i.e. taking the actual simulated rotor-average wind speed and direction and turbulence intensity as if they were the measured values. The values

were low-pass-filtered using a first-order filter with a time constant of 60s to represent approximately the way in which these signals would be derived in the field. Further filtering could be done, for example to help represent advection of the wind conditions along the line of turbines, but a systematic study was not conducted as this option was not available in the farm control software implemented in the field.

## 4.5 Simulation results with setpoint smoothing

Site met mast data with suitable wind conditions for a period of just over 5 hours was selected, and used to generate a wind field covering the 9-turbine row. The simulation wind conditions are illustrated in Figure 7.

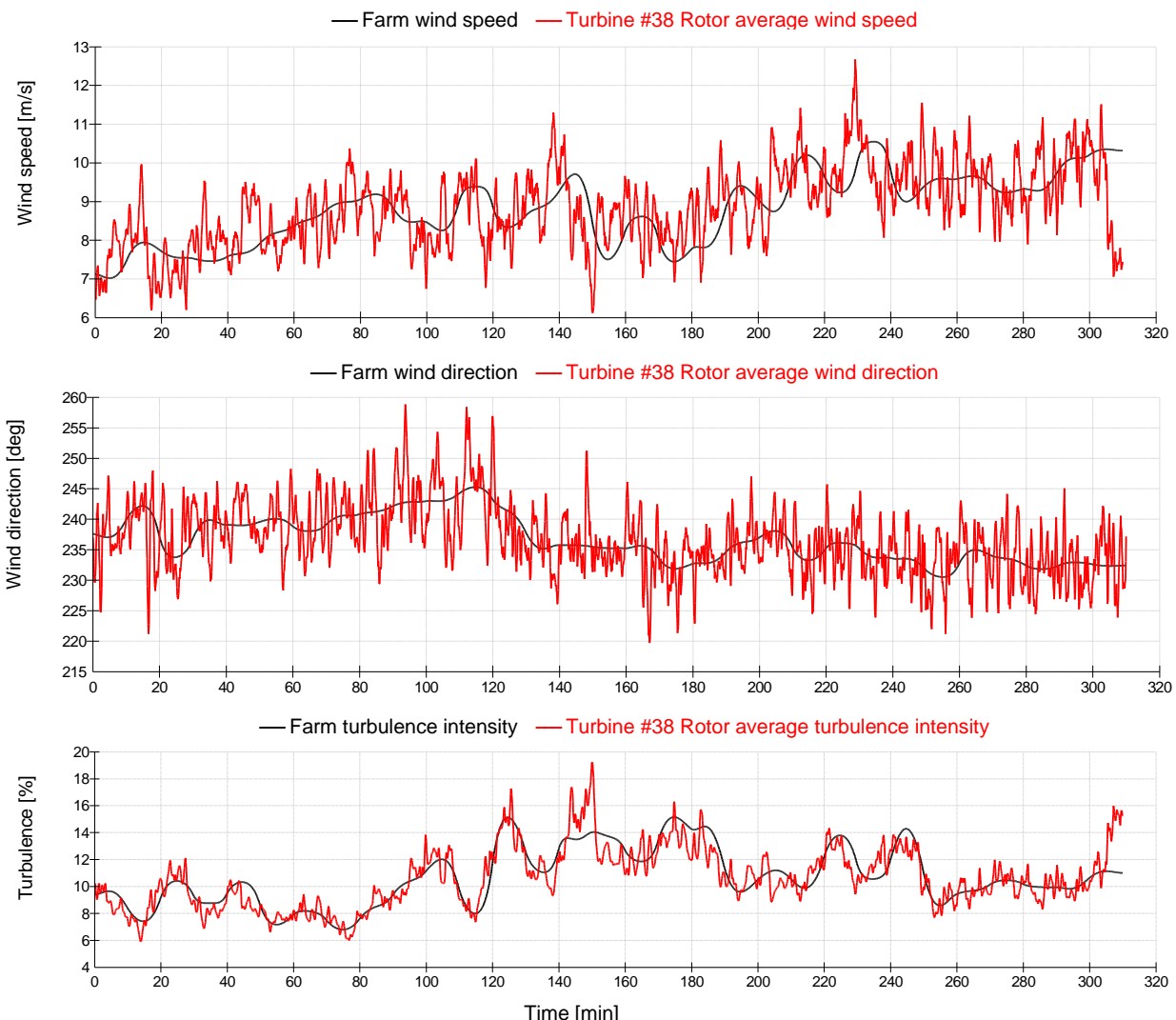

**Figure 7: Wind conditions for the initial simulations. The black line represents the smoothed 10-minute mast data which is assumed to apply at a point halfway down the row of turbines. The red line shows conditions from the simulated wind field at the turbine #38 rotor.**

Using this wind field, four simulations were carried out:

- Base case, without induction control
- Induction control, using the raw optimised setpoints
- Induction control, with the setpoint table smoothed to account for a 5° uncertainty in wind direction

- Induction control, with the setpoint table smoothed to account for uncertainties of 5° in wind direction and 1m/s in wind speed

Figure 8 shows how the setpoint variation becomes much smoother, using the first controlled turbine (#37) as an example. The effect on the total power production from the nine turbines, shown in Figure 9, is difficult to discern in the plot, so the mean values are given in Table 1.

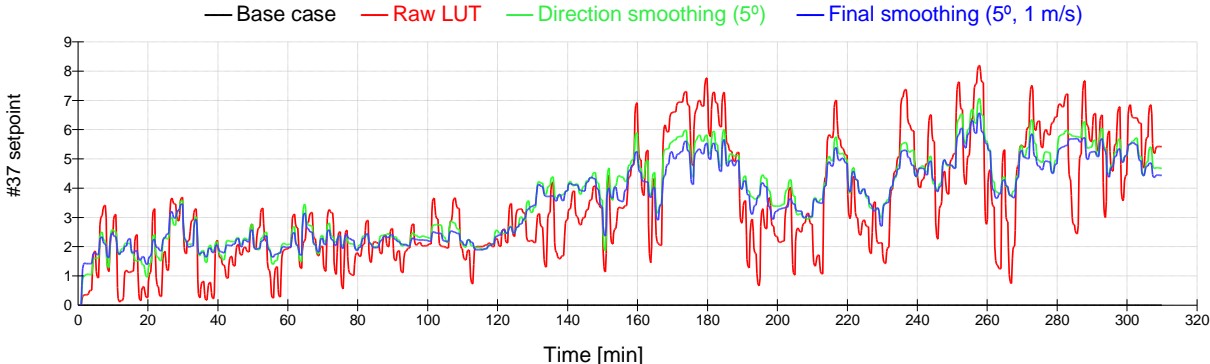

**Figure 8: Effect of LUT smoothing on induction control setpoints (turbine #37 illustrated). For the base case, the setpoint is zero.**

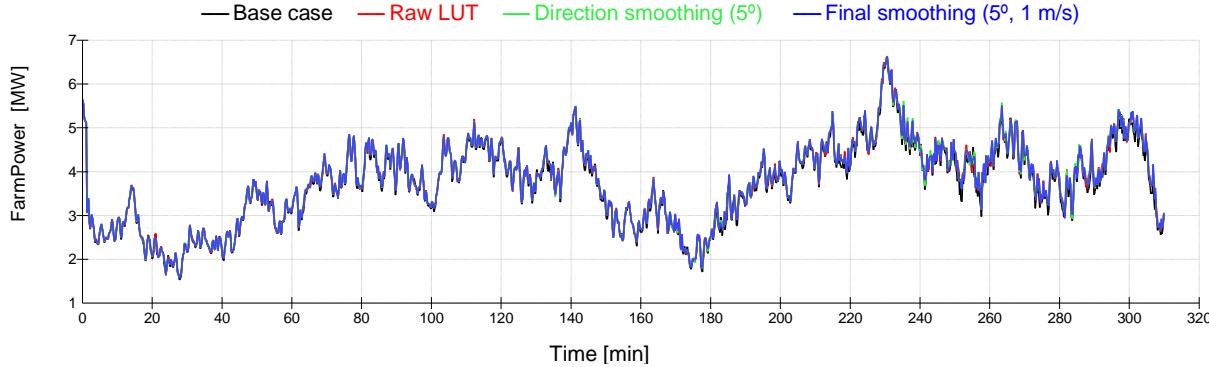

**Figure 9: Effect of LUT smoothing on total power output (note the plots are almost indistinguishable)**

| Case | Power [MW] | Increase [%] |
|---|---|---|
| Base case | 3.7058 | 0 |
| Raw LUT | 3.7613 | 1.50% |
| Direction smoothing (5°) | 3.7641 | 1.57% |
| Final smoothing (5°, 1m/s) | 3.7645 | 1.58% |

**Table 1: Mean power values from Figure 9**

As well as giving smoother control action, this shows that smoothing to account for wind uncertainties, especially wind direction, increases the power gain achieved by induction control. This smoothing was therefore adopted for the LUT used in the field tests. More simulations could be run to optimise the amount of smoothing, but this was not considered worthwhile at this stage.

**4.6 Simulation of controller toggling**

As a final test prior to the start of field testing, a longer simulation was run using a different sample of met mast data to generate the wind field, this time 22.5 hours in length, shown in Figure 10.


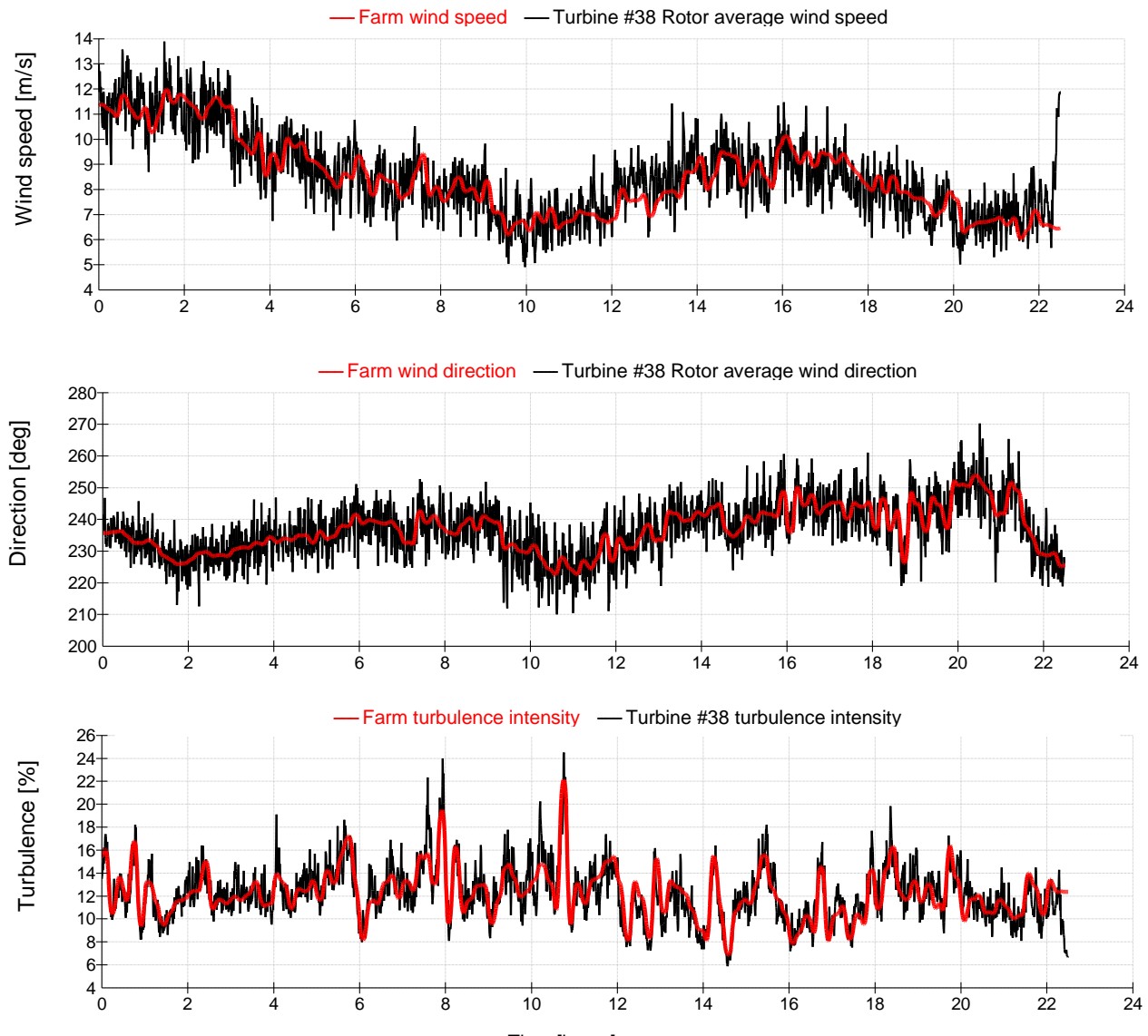

**Figure 10: Wind conditions for the toggling simulations. The red line represents the smoothed 10-minute mast data which is assumed to apply at a point halfway down the row of turbines. The black line shows conditions from the simulated wind field at the turbine #38 rotor.**


Three simulation were run using this wind field:

- Base case, without induction control
- Induction control with the final smoothed LUT
- Induction control toggling on and off every 35 minutes, as for the field tests

Figure 11 shows the power reduction setpoint at the first controlled turbine (#37), demonstrating the toggling effect in the third simulation.

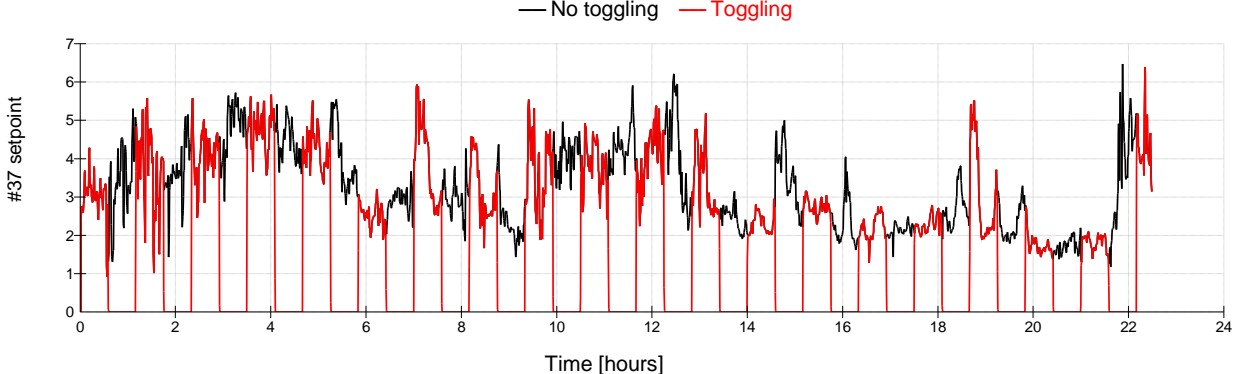

**Figure 11: Induction control setpoints showing controller toggling (turbine #37 illustrated)**

The total power for the nine turbines is shown in Figure 12 for all three simulations. The difference is difficult to discern in
the plot, so the mean values are given in Table 2. For this period, the induction control increases the power output by 1.3%, and if toggling on and off, this increase is halved, as would be expected.

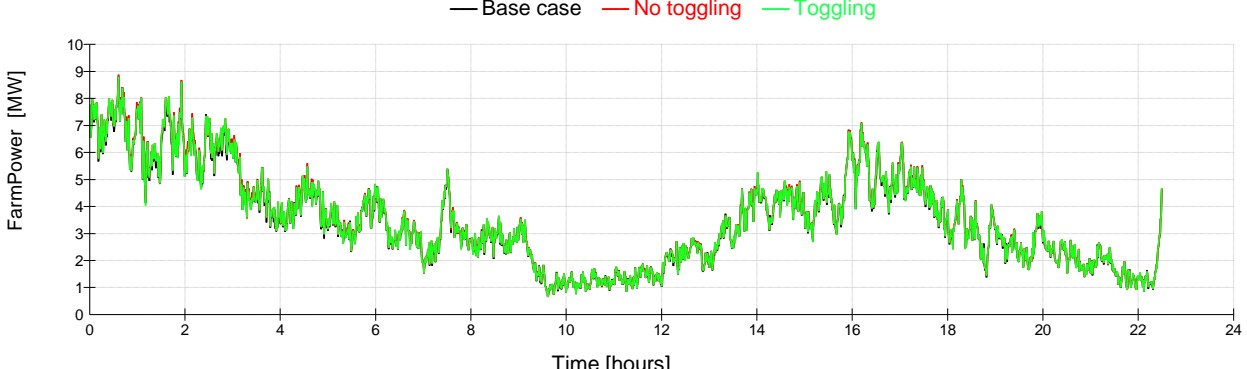

**Figure 12: Total power output for the toggle test simulations**

| Case | Power [MW] | Increase [%] |
|---|---|---|
| Base case | 3.496 | 0 |
| Induction control | 3.541 | 1.29% |
| Induction control toggled on and off | 3.519 | 0.65% |

340                                           **Table 2: Mean power values from Figure 12**

## 5 Field testing

The induction control test was initiated on site and data recording started on 11<sup>th</sup> July 2019. The following day, an offset applied to the wind direction used for the LUT, obtained empirically by matching measured directions to the directions where maximum wake deficits were observed, was corrected, so valid SCADA data was available from 10:50 on 12<sup>th</sup> July onwards. The SCADA data was recorded with a one-minute sampling frequency, and provided in a Matlab datafile. The file was updated periodically to include the latest data, which was analysed as described below. Some apparent inconsistencies were checked by running simulations with LongSim using wind fields created from the actual Turbine #38 SCADA data, and with setpoints toggled according to a flag recorded in the SCADA data, to try to mimic as closely as possible what was happening in the field. Comparison of simulated and measured results for all the turbines proved extremely useful, and revealed some interesting inconsistencies. For example, Figure 13 compares the simulated and measured power at turbines #34 and #33 during a 17-hour period. The power is very well predicted for #34, and similarly for all the other turbines except for #33: it is clear that this turbine was running in a curtailed mode. Unfortunately, the status flags in the recorded SCADA data did not include any indicator of curtailment.

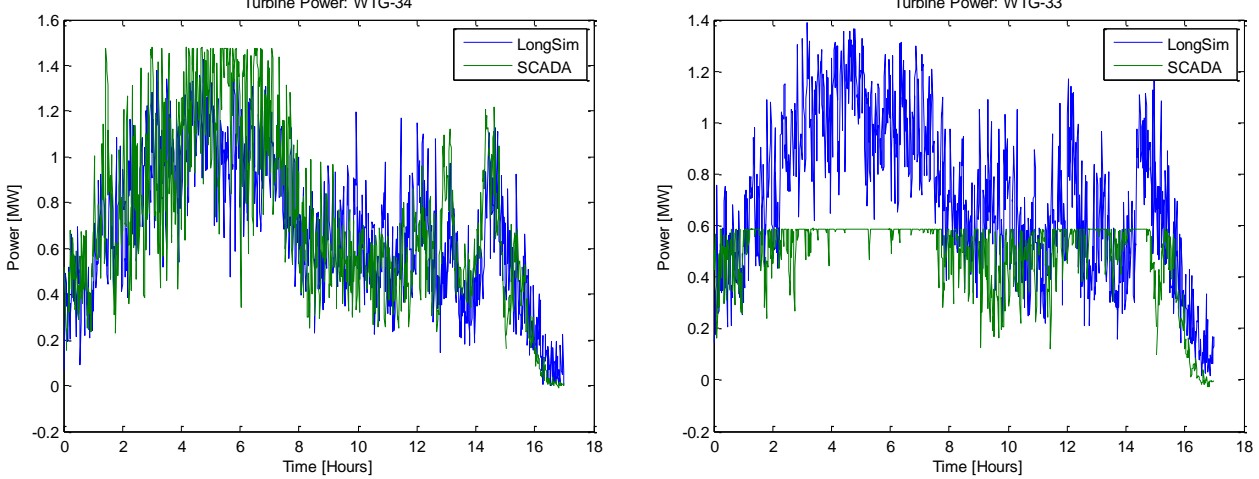

**Figure 13: Measured and simulated power at turbines #34 and #33**

These simulations also proved to be a valuable tool for verifying the correct implementation of the setpoint changes in the field, as the simulated and measured setpoints for any turbine should match fairly closely through the period of the simulation. Figure 14, for example, shows an excellent match, and any significant discrepancies could be easily identified.

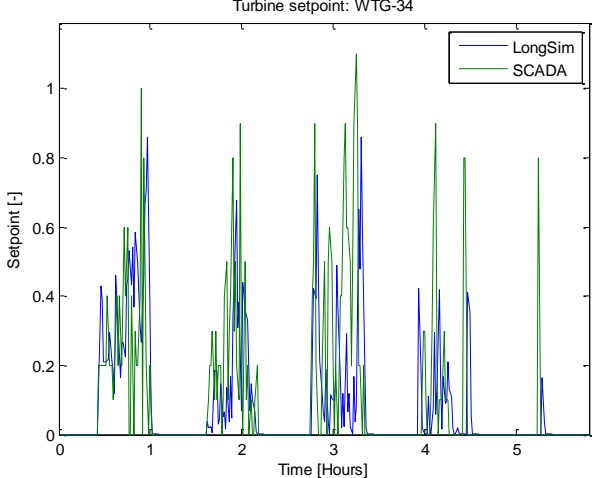

**Figure 14: Measured and simulated setpoints at turbine #34**

### 5.1 Analysis of field test data

The final dataset consisted of more than 6 months of SCADA data for the nine turbines at 1-minute resolution (298066 records). This was run through an analysis program which carried out the following steps:

1. The data was filtered to include only those records which were both relevant and correct. Firstly, some 43% of the records were rejected because of missing values of any of the variables of interest, namely the time stamp, the power at each turbine, the wind conditions (speed, direction and turbulence intensity) used for the setpoint table lookup, the operational state of each turbine, and the controller toggle state. Any records from periods when there were known technical issues affecting the control states were also discarded. The valid records were then filtered to include only the relevant wind conditions for which the control is active, namely wind speed in the range 6 – 15 m/s and direction in the range 180 – 270 degrees, as no setpoints were applied outside of this range. This left 21965 relevant records. Records with high or low turbulence intensity were not filtered out, because the setpoints continued to be applied even if the turbulence was out of the range for which they were designed. Finally, records where one or more turbines were not in normal operation were also discarded, leaving 12498 records, or just over 4% of the original data. For the sake of the subsequent processing steps, rather than actually discarding any records, the filtering was done by assigning a logical flag to each of the 1-minute records to say whether or not that record is accepted.

2. The data is parsed to find the moments at which the toggle flag changes. The 5 minutes following the toggle change are discarded as 'settling time', and following this, 10-minute chunks are collected up to the next toggle change. Since the toggle interval is 35 minutes, there should be three such 10-minute chunks in each toggle period. However, the realities of real life mean that this is not always exactly true, so a 10-minute chunk is kept as long as its apparent length defined by the recorded start and end time is within 30 seconds of 10 minutes.

3. For each 10-minute chunk, the mean value of the filter flag is calculated, and the chunk is accepted if this is greater than 0.9 (i.e. at least 90% of the points within it are accepted). For each such chunk, the mean power (summed over turbines) is calculated, as well as the mean lookup table wind speed, wind direction and turbulence intensity, and also the mean toggle state (control 'ON' or 'OFF'). The mean normalised power is also calculated, defined as the total power from the 9 turbines divided by the power at the reference turbine #38. Each chunk is classified as having control 'ON' if the mean toggle state is greater than 0.9, or 'OFF' if less than 0.1 (these criteria are only needed to cope with occasional irregularities in the data).

4. The 10-minute 'ON' and 'OFF' chunks are then binned according to wind conditions.

## 5.2 Field test results

The top left graph in Figure 15 shows the mean 'ON' and 'OFF' power in each wind speed bin. The crosses show the standard deviations of the points in the bin, and the bar chart below shows the number of 'ON' and 'OFF' points in each bin. There appears to be a consistent increase over the wind speed range of interest, apart from the 6-7 m/s bin, although it should be noted that the increase is generally smaller than the standard deviation of the points. The highest wind speed bin does not have enough points to be meaningful. Note also that at the lowest wind speeds, some heavily waked turbines may not be producing any power, and in that situation, the thrust coefficient depends on the supervisory control – a turbine generating no power might continue to rotate at minimum operating speed, or it might slow down to an idling speed, probably depending on how long the power remains low. No information was provided about this, so the setpoint optimisations assumed an intermediate thrust coefficient of 0.3 for any turbine producing zero power. This represents a source of uncertainty at the lowest wind speeds. The 'unweighted increase' figure simply represents the increase in the sum of the mean powers in all bins containing at least two 'ON' and two 'OFF' points, i.e. excluding the highest bin in this case. The lower plot shows the average turbulence intensities in each bin. These are all higher than the maximum 17% turbulence for which the controller was designed, and according to the modelling, the control performance decreases at higher turbulence. In many bins, the average turbulence intensity of ON points happens to be slightly higher than for OFF points, so it is possible that higher wake dissipation rather than the control action might account for some of the power increase. It is unclear how much of the 'unweighted increase' of 2.3% is due to higher turbulence intensity and how much is due to the control.

The right hand side of Figure 15 shows the points binned against wind direction. Since the points in any bin might all have significantly different wind speeds, it makes sense to plot the mean normalised power as defined above, rather than the mean absolute power. Again, as there are not very many points per bin, the increase is smaller than the standard deviations, but the increase seems consistent. However, the largest increases are in the first few bins, where wake interactions (and hence the benefits of the control) should be small, but these points are unreliable because there are very few ON points, and with particularly high turbulence intensities. For the bins above 220 degrees, there are reasonable numbers of points and the ON

and OFF turbulence intensities are very similar (though still well above 17%), so the power increase starts to be credible. The unweighted increase is calculated as before – in this case all bins have enough points to be included, but the figure is

clearly skewed by the unreliable increases in the first few bins. One would expect the two unweighted increase figures to converge once all the speed and direction bins are fully populated, since they represent the same set of datapoints.

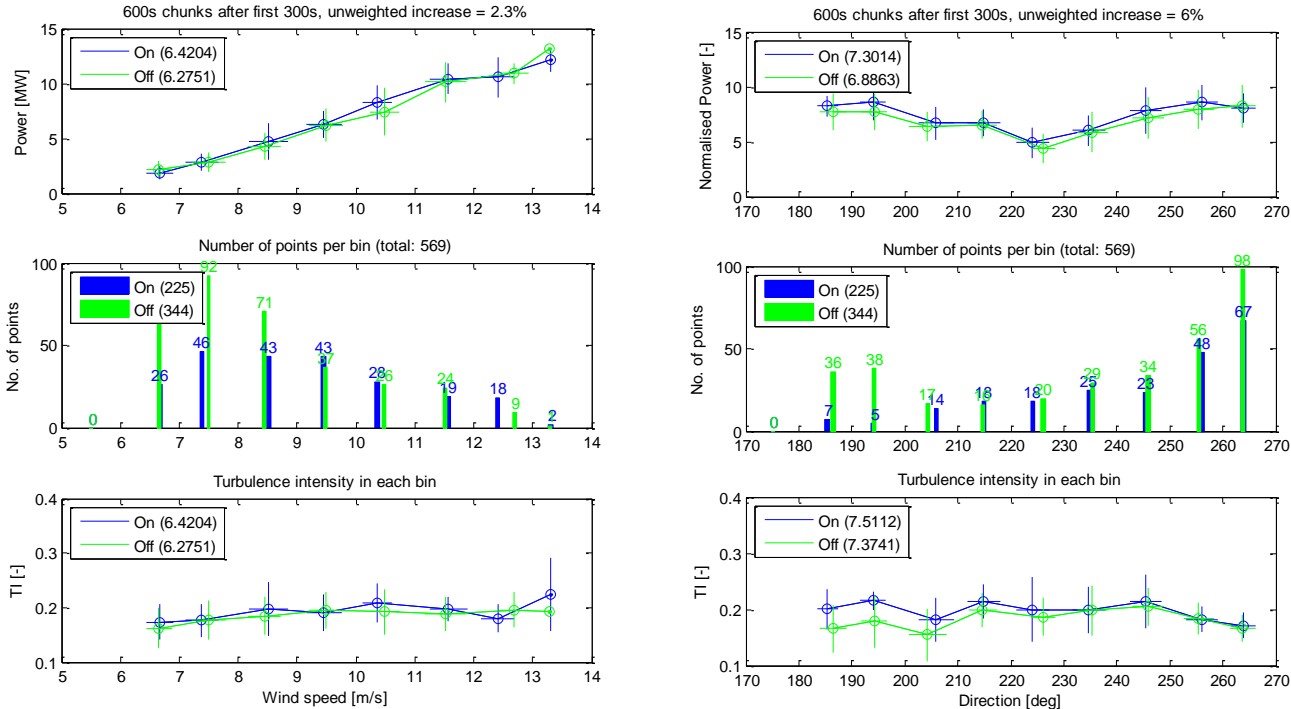

**Figure 15: Field test results binned on wind speed and direction**

To better understand these results and how they relate to the model predictions, the points would need to be binned in three

dimensions against wind speed, wind direction and turbulence intensity. There are clearly not enough points for this, but some insights can still be gained by two-dimensional binning on speed and direction. Figure 16 shows the ratio of mean ON and OFF power in each speed and direction bin containing at least one ON and one OFF point. The ratios are mostly greater than 1, peaking at 1.71 in just one bin, i.e. an increase of 71%. This extreme value is clearly not credible, but must be seen in the context of the actual numbers of points in each bin, shown in Figure 17, and the turbulence intensities shown in Figure

18. The bin with the 71% increase contains just 4 ON points and 7 OFF points, and the average turbulence for the ON points is significantly higher. Most bins contain even fewer points, and in some bins the power ratio is less than 1. Many of the points (even more OFF points) are concentrated at low wind speed with directions above 260 degrees, which is right at the edge of the region where induction control is expected to be useful. The mean increase over all bins containing at least one ON and one OFF point is shown in Figure 16 as 2.42% over 47 bins. If we only accept bins with at least two ON and two

OFF points, the mean increase is 4.66% over 33 bins, and if we require at least three points, it is 4.97% over 21 bins (note

that this includes some valid bins which do not show up in the contour plots because they are isolated from neighbouring bins).

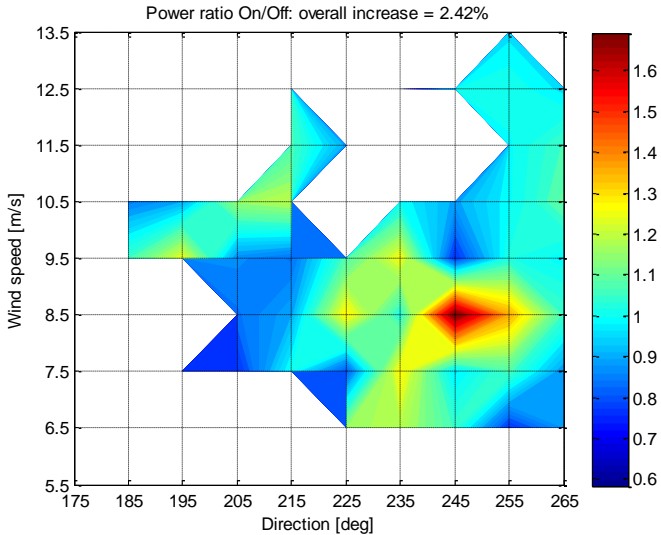

**Figure 16: Power ratio binned on wind speed and direction**

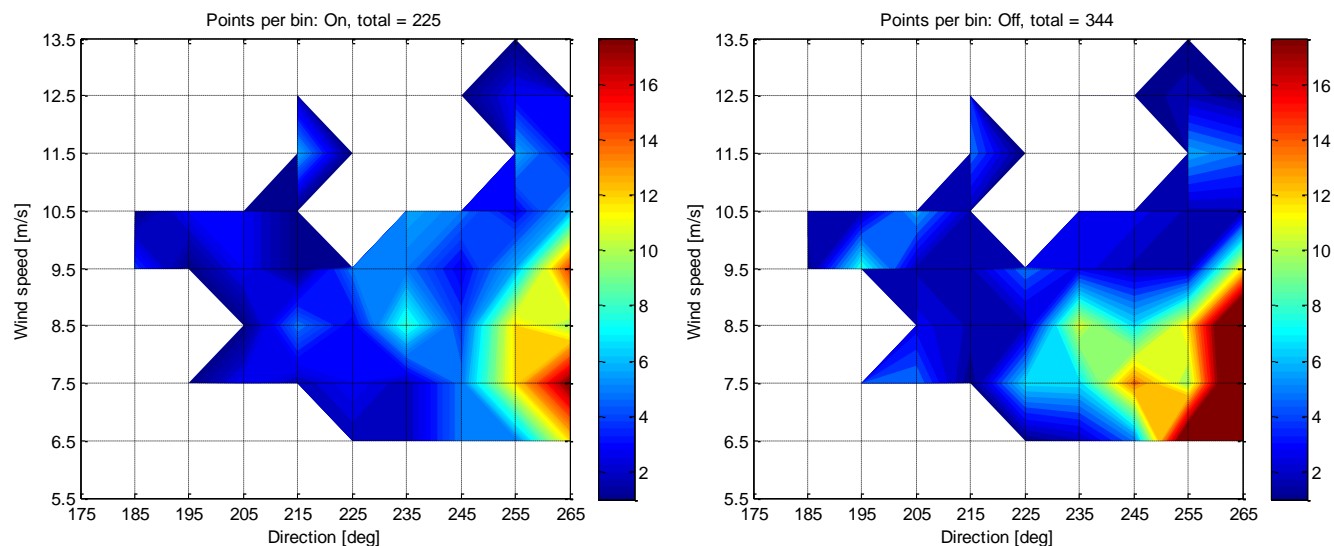

**Figure 17: Numbers of points in each bin**

Also, from Figure 18, which shows the mean turbulence intensity for the ON and OFF points in each of the bins, it is clear that higher turbulence intensities were experienced during most of the measurement period than the 17% maximum that the induction control was designed for. The induction control is expected to be less effective in higher turbulence intensities, due to faster wake dissipation.

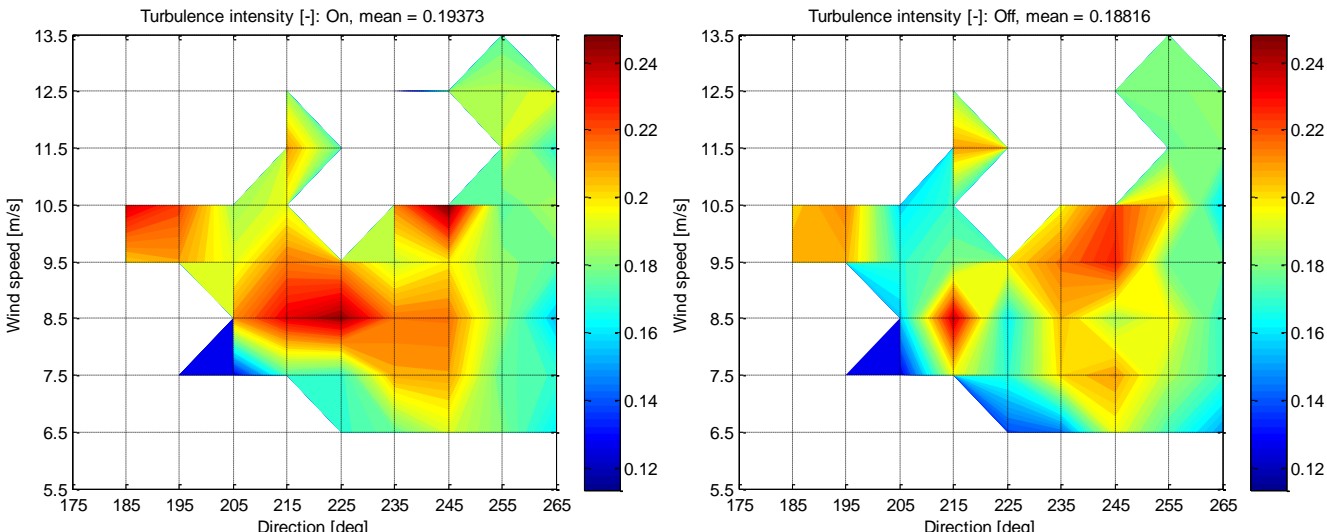

**Figure 18: Mean turbulence intensities in each bin**

There is not enough data to bin in three dimensions, but to try to better understand the effect of turbulence intensity, the same data can be binned on direction and turbulence intensity, this time binning the ON/OFF ratio of the normalized power (using

the power of turbine #38 as reference) to remove effect of different wind speeds within each bin. The results are shown in Figure 19, together with the corresponding plots showing the number of points per bin and, now, the mean wind speed per bin.

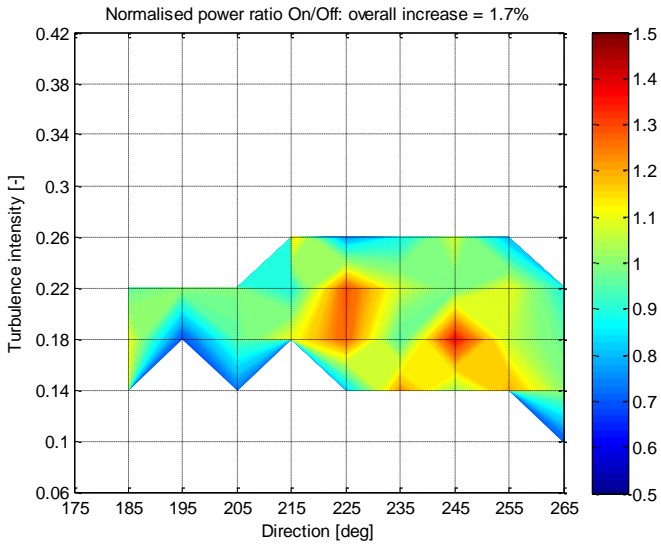

**Figure 19 (continued on next page)**

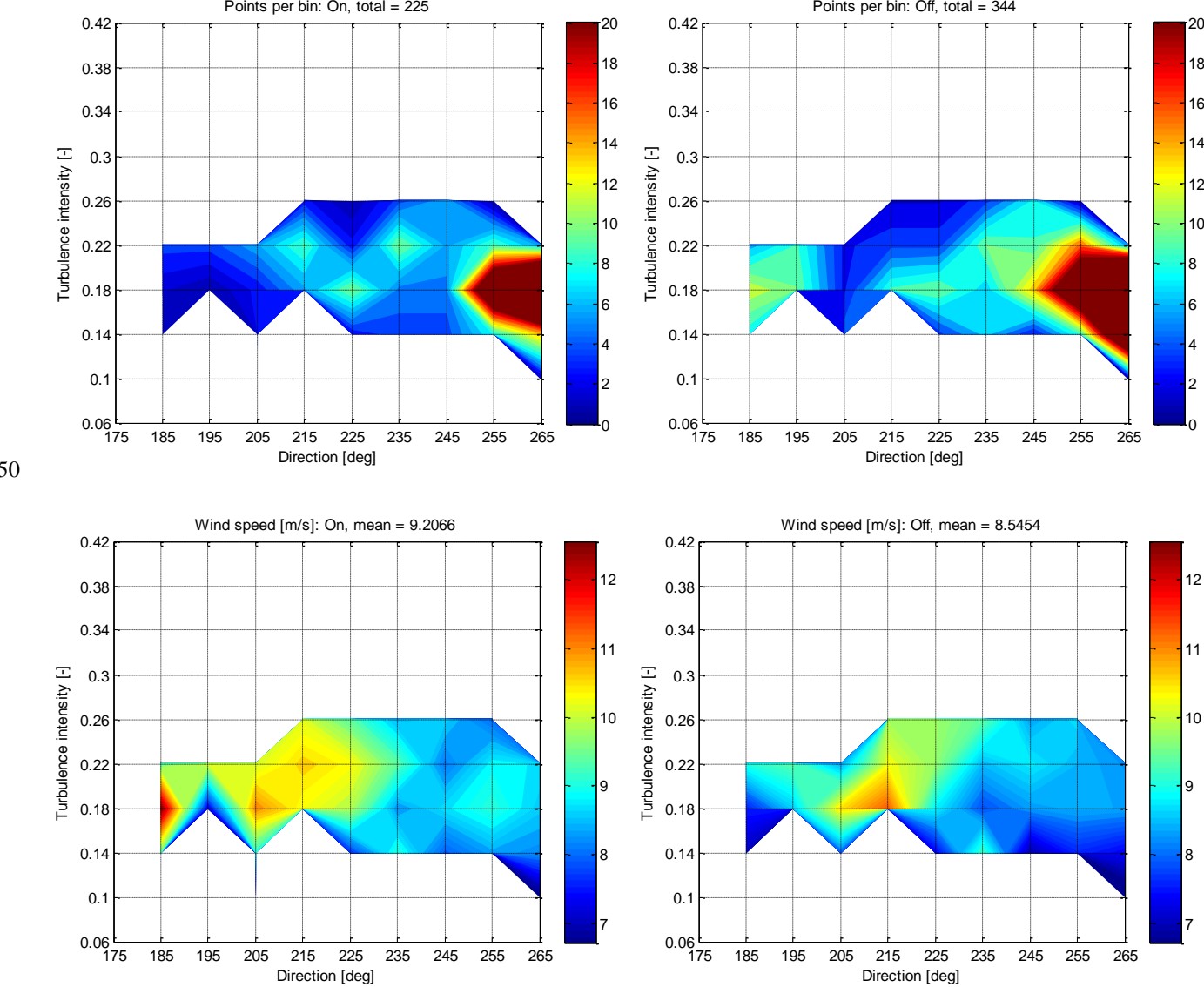


**Figure 19 (concluded): Normalised power ratio binned on turbulence intensity and direction**

Now we can see that in many of the populated bins, there is a generally positive increase, which exceeds 20% in five bins, and averages 1.7% overall. The full statistics of the populated bins in Figure 19 are listed in Table 3, including the mean
wind speed and turbulence intensity of the ON and OFF points. The number of points in each bin is not large, as expected. The turbulence intensities are necessarily quite similar within each turbulence bin; the mean wind speeds differ significantly across the analysed bins, but any potential bias due to wind speed variation is mitigated by the use of normalised power. Overall, there is no clear evidence for any bias arising from chance variations with this small number of points.

| Power Ratio | Direction bin [deg] | Turbulence bin [-] | Points in bin ON | Points in bin OFF | Mean wind speed ON | Mean wind speed OFF | Mean turbulence ON | Mean turbulence OFF |
|---|---|---|---|---|---|---|---|---|
| 1.37 | 245 | 0.18 | 5 | 14 | 8.77 | 8.31 | 0.1833 | 0.1856 |
| 1.30 | 225 | 0.22 | 3 | 4 | 10.39 | 9.79 | 0.2056 | 0.2191 |
| 1.28 | 225 | 0.18 | 10 | 10 | 9.96 | 8.85 | 0.1879 | 0.1756 |
| 1.23 | 235 | 0.14 | 4 | 7 | 9.10 | 9.29 | 0.1559 | 0.1380 |
| 1.21 | 255 | 0.14 | 8 | 9 | 8.42 | 7.68 | 0.1527 | 0.1462 |
| 1.19 | 205 | 0.10 | 2 | 6 | 7.17 | 9.03 | 0.1190 | 0.1151 |
| 1.17 | 215 | 0.26 | 3 | 2 | 10.18 | 9.88 | 0.2612 | 0.2483 |
| 1.13 | 185 | 0.14 | 1 | 8 | 9.59 | 7.09 | 0.1491 | 0.1422 |
| 1.11 | 185 | 0.18 | 2 | 12 | 12.66 | 7.62 | 0.1721 | 0.1812 |
| 1.11 | 215 | 0.18 | 6 | 9 | 10.34 | 11.33 | 0.1839 | 0.1829 |
| 1.10 | 255 | 0.22 | 9 | 16 | 8.89 | 8.47 | 0.2128 | 0.2163 |
| 1.07 | 245 | 0.26 | 6 | 8 | 8.65 | 8.38 | 0.2530 | 0.2516 |
| 1.07 | 235 | 0.22 | 10 | 10 | 9.45 | 8.49 | 0.2206 | 0.2189 |
| 1.06 | 255 | 0.18 | 30 | 30 | 9.15 | 8.41 | 0.1790 | 0.1752 |
| 1.03 | 245 | 0.14 | 4 | 3 | 7.95 | 6.97 | 0.1404 | 0.1462 |
| 1.02 | 265 | 0.14 | 14 | 32 | 8.29 | 7.43 | 0.1464 | 0.1415 |
| 1.01 | 195 | 0.22 | 4 | 9 | 10.14 | 9.41 | 0.2205 | 0.2167 |
| 0.99 | 205 | 0.22 | 6 | 3 | 10.13 | 8.40 | 0.2159 | 0.2320 |
| 0.99 | 245 | 0.22 | 6 | 9 | 8.17 | 8.80 | 0.2232 | 0.2147 |
| 0.97 | 265 | 0.18 | 45 | 58 | 8.63 | 8.36 | 0.1766 | 0.1785 |
| 0.96 | 185 | 0.22 | 4 | 6 | 9.55 | 9.07 | 0.2270 | 0.2130 |
| 0.96 | 205 | 0.18 | 3 | 3 | 11.07 | 10.51 | 0.1903 | 0.1925 |
| 0.95 | 235 | 0.18 | 5 | 7 | 8.33 | 7.94 | 0.1806 | 0.1852 |
| 0.90 | 265 | 0.22 | 6 | 6 | 8.80 | 8.26 | 0.2090 | 0.2080 |
| 0.89 | 215 | 0.22 | 9 | 4 | 10.65 | 10.29 | 0.2183 | 0.2223 |
| 0.89 | 235 | 0.26 | 4 | 4 | 8.52 | 9.45 | 0.2515 | 0.2577 |
| 0.84 | 225 | 0.14 | 3 | 4 | 7.74 | 7.43 | 0.1480 | 0.1508 |
| 0.81 | 255 | 0.26 | 1 | 1 | 8.07 | 8.72 | 0.2490 | 0.2489 |
| 0.77 | 225 | 0.26 | 1 | 2 | 9.46 | 9.83 | 0.2410 | 0.2480 |
| 0.75 | 205 | 0.14 | 3 | 5 | 9.19 | 8.05 | 0.1469 | 0.1343 |
| 0.71 | 265 | 0.10 | 2 | 2 | 6.73 | 6.47 | 0.1166 | 0.1102 |
| 0.69 | 195 | 0.18 | 1 | 10 | 7.36 | 8.62 | 0.1944 | 0.1891 |

**Table 3: Statistics of populated bins in Figure 19 (ordered by the normalised power ratio)**

 **5.3 Model validation using field test results**

Finally, the field test results have been used to validate the LongSim model, by running the model in conditions matched to the field test conditions as closely as possible, both in the steady state and in dynamic simulations.

### 5.3.1 Steady-state model validation

The model was run in steady state, both with and without induction control, for each of the bins containing at least one ON
and one OFF point, corresponding to Figure 16. For each bin, the mean wind conditions (speed, direction and turbulence intensity) for the ON points was used as input to the model with induction control on, and the mean conditions for the OFF points were likewise used for the model runs with no induction control. The results are shown in Figure 20, which should be compared against Figure 16. The predicted overall increase, 2.38% is very similar to the field test result of 2.42%. However there are differences in individual bins. There is a very similar peak increase of 68% at 8.5 m/s, but at a different direction:
225 compared to 245 degrees. In the measured data, the 8.5 m/s 225 degree bin showed a 29% increase, but it contained only 3 ON and 2 OFF points. For the 8.5 m/s 245 degree bin, the model predicts a 17% increase rather than the measured 71%. The model also predicted a large 80% increase in the 6.5% 235 degree bin, but in this situation there would be some waked turbines generating zero power, and the assumed thrust coefficient may not be correct, as mentioned above.

Apart from these differences in specific bins, and bearing in mind the small numbers of measured points in most bins, the
general pattern of results over most of the bins indicates a quite encouraging comparison between modelled and measured results.

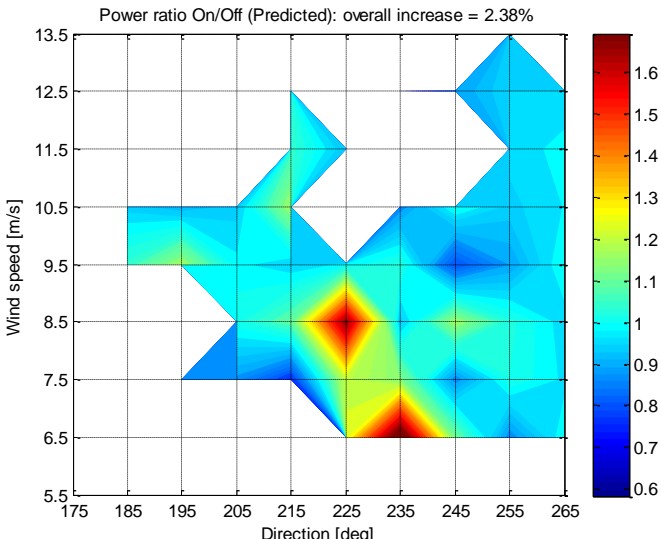

**Figure 20: Ratio of model predictions of power in each bin (compare Figure 16)**

### 5.3.2 Dynamic model validation

For the dynamic model validation, a period of just under 20 hours (06-Dec-2019 13:25:00 to 07-Dec-2019 09:14:00) was selected for which the wind conditions were appropriate for a reasonable amount of induction control activity to take place. A wind field was generated from the SCADA wind conditions as explained in Section 4, and used as input to a LongSim simulation, and using the SCADA toggle flag to switch the control on and off. The simulated turbine power and setpoint time histories were then compared against the measured SCADA data. The wind conditions are shown in Figure 21. These

conditions are applied at a point close to the middle of the turbine row. For all other points, a wind field is generated stochastically by LongSim using a random number generator with assumed spectrum and coherence functions, so while the simulated results are expected to match the measured data at low frequencies, the higher frequency 'noise' should only have similar statistical characteristics rather than matching exactly second by second.

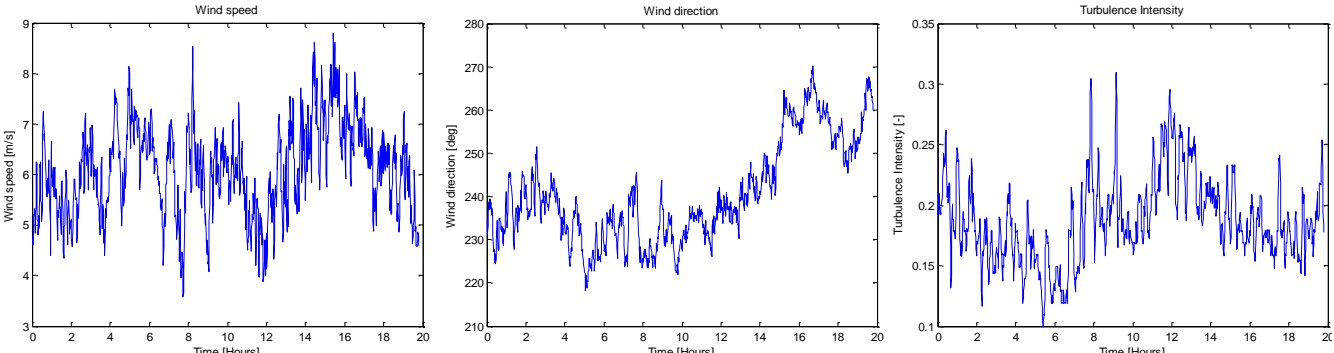

**Figure 21: SCADA wind speed, direction and turbulence intensity used for simulation**

Figure 22 shows the power production at the first three turbines, and also the last turbine. The power at turbine #38 is very well predicted. At turbine #37, it appears that the turbine must have been switched off for about the first 3.5 hours, but the agreement after that is very good. At the next turbine, #36, the measured power is higher than predicted by LongSim for the first 3.5 hours, presumably due to the fact that while #37 was not generating, it was not waking #36, whereas the simulation

was not aware of the curtailment. After #37 started generating, the agreement is again very good. There is good agreement for the other turbines too, suggesting that wake effects are well predicted all along the row. Even for turbine 13, the agreement is quite good although it is a long way from where the wind speed used for the simulation was measured. Terrain effects on wind speed have not been modelled, and discrepancies at the higher frequencies are expected because the higher frequencies in the simulated wind field are synthesised statistically.

Figure 23 shows the toggling power reduction setpoints at the first four controlled turbines. With the usual exception of the first 3.5 hours for turbine #37 when it was curtailed, the agreement is again very good. This is equally true for the three other controlled turbines, not shown.

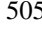

**Figure 22: Measured and predicted power at turbines #38, #37, #36 and #13, and the total of the nine turbines**

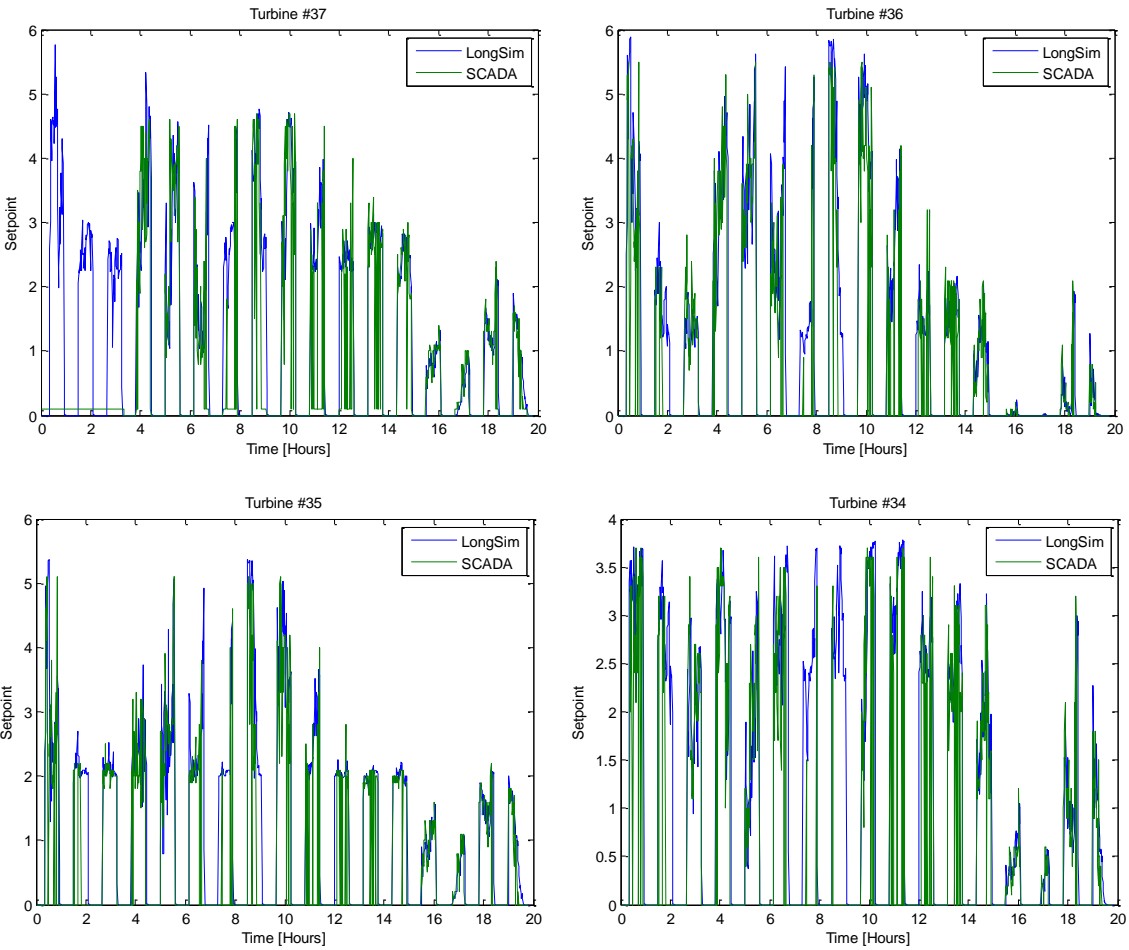

**Figure 23: Measured and predicted setpoints at turbines #37, #36, #35 and #34**

**Conclusions**

As part of the EU Horizon 2020 research project CL-Windcon, a field test of an axial induction controller for a row of nine turbines at Sedini wind farm in Sardinia, Italy was carried out. The aim of the controller was to reduce individual turbine setpoints as a function of wind conditions, so as to reduce wake losses and increase the overall power output from the whole row. Historical data from the site was first used to confirm a choice of wake model, and the optimiser of the LongSim model was then used to generate turbine setpoint lookup tables as a function of wind speed, direction and turbulence intensity which would maximise the power output from the row. The tables were then incorporated into a practically realisable control algorithm, which makes use of available measurements to estimate the wind conditions and takes account of wind speed and

direction uncertainties. Using wind inputs derived from historical site data, dynamic time-domain simulations were performed in LongSim to verify the design choices and predict the likely dynamic performance.

The algorithm was then implemented in the field, and data was collected for over six months, with the control action toggling on and off at 35-minute intervals so that the effect of the controller could be assessed. Because of the low occurrence of the appropriate wind conditions, and after filtering out any invalid records, there were eventually about 200 hours of useful data, from which about 570 ten-minute periods could be extracted, covering a range of wind conditions. This number of datapoints was too small to be able to quantify the improvement precisely in a statistically meaningful way, and much too small to allow the data to be binned against all three of the most relevant variables (wind speed, wind direction and turbulence intensity). Alternative ways to bin the data against one or two variables at a time were therefore used to help identify possible biases, such as might be caused by differences in turbulence intensity within a bin, and normalising the power by the power of the leading turbine was useful to compensate for differences in wind speed. Alternative binning methods resulted in estimates in the range 1.7% to 2.4% for the average power increase over the relevant range of wind conditions, although it remains uncertain how much of this might still be attributable to other factors such as turbulence intensity. In our view, at least a few months of valid data would be required to achieve a reasonable level of confidence.

Furthermore, the measured data was also used for validation of the LongSim software, demonstrating excellent agreement and confirming the suitability of LongSim as a valuable tool for designing and testing wind farm controllers.

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

**Author contribution**

Ervin Bossanyi developed the wind farm model, designed the controller and analysed the test results. Renzo Ruisi performed the site atmospheric stability analysis and the wake modelling analysis using historical SCADA data.

**Competing interests**

The authors declare that they have no conflict of interest.

**Acknowledgements**

The authors would like to acknowledge the excellent cooperation of Stefan Kern and his team from GE Renewable Energy and Giancarlo Potenza from ENEL Green Power. The field testing described in this paper was only possible with their dedicated professionalism and the hard work of their teams, both on and off site.

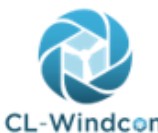 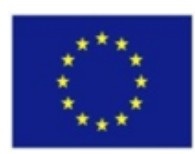


This project has received funding from the European Union's Horizon 2020 research and innovation programme under grant agreement No 727477