# Peer review of "Axial induction controller field test at Sedini wind farm"

_Wind Energy Science, 2020_

## Short Comment (SC1) · 8 Jul 2020

Dear authors,

Thank you for this very interesting contribution, it was a pleasure reading it! I have two comments that might be worth considering to strengthen the conclusions and credibility of the experimental results:

1) P20/L386: "If we only accept bins with at least two ON and two OFF points..." It would be interesting to see how the mean increase changes when increasing the minimum number of points per bin (further than three). It seems from the figures shown that the average increase might become smaller when considering the more trustworthy bins only (for example where more than five data points are available). Also, it might

be interesting to report a weighted mean that takes the number of occurrences into account.

2) P22/L394: "So far, no account has been taken of turbulence intensity in the data analysis." The TI has a known effect on the wake recovery and meandering. Therefore, higher turbulence might also lead to increased farm power production. Indeed, the average TI during farm control being "ON" is larger than during "OFF", especially in the bin with a 71% power increase. Could it be that the reported power increase is partially caused by increased TI? Is it possible to expand Figure 14 and include the TI (with normalized power as it is done for the wind direction)?

Additional comment: Figure 7 shows the induction control setpoints (ranging between "0" and "9"). I could not find any definition for those setpoints. Does "9" stand for 90% power curtailment?

Thank you for your consideration!

---

## Author Comment (AC1) · 14 Jul 2020

Thank you for the useful and relevant comments. I would like to respond as follows:

1) Ideally we would indeed increase the minimum acceptable number of points per bin, as you suggest. The problem here, because we have a very limited amount of data, is that when binning by both wind speed and direction, the number of points per bin is very small, and if we increase the threshold number of points, we accept even fewer bins. This might give more confidence in the calculated increase, but it also changes its meaning since the range of wind conditions covered will be different. The mean increase over those bins for which we happen to have enough data is not a particularly meaningful number anyway – an estimate of the annual average increase would be

[Figure]

more interesting, but clearly we don't have enough data to fill all the bins which would be needed for this. We preferred to simply present the data as is, including the points per bin, and we hope you agree that the words in the conclusions of the paper give a fair assessment of the outcome of the experiment.

2) You are right that in addition to the small number of points in the bin, TI variations may help to explain the implausibly large increase in that one bin. Thank you for noticing that – we propose to add a comment to that effect. Your suggestion to extend figure 14 is also a good one – we can show the mean TI for the ON and OFF cases for each bin.

3) You also ask about the meaning of the setpoint values in figure 7. The setpoint ranges from level 0 (normal operation) to level 10 (maximum power reduction). For each level, the power reduction is not fixed but depends on the wind speed, having no effect in low and high winds. Details of the algorithm used to reduce the power were not provided by the manufacturer for reasons of confidentiality, but the maximum reduction does not exceed 20% of rated power.

---

## Referee Comment (RC1) · Bart M. Doekemeijer (Referee) · 5 Aug 2020

**Review WES-2020-88**

Dear authors,

I enjoyed reading your article on the field experiment at Sedini investigating axial induction control. I find this work to be extremely important in the further validation of wind farm control technologies. The article contains a lot of information and generally has a good story. Though, I have a number of major comments on the article. I want to stress that these comments are my opinion, and with the sole intention of improving the scientific relevance of the manuscript. These comments are not meant to discourage the authors in their work, and I want to again emphasize that I am convinced of the article's importance. I hope this feedback finds you well.

**General comments:**

1. The core results of this article, based on the title and my interpretation of the article, would be represented by Figures 15 and 16. These figures show the reported field experiment gains in power production using axial induction control compared to baseline operation. The authors show gains of up to 70% and losses of up to 40% using this algorithm. Based on what the literature has shown before on axial induction control, I find such values difficult to believe; I would expect the gains/losses to be in the order of 5% at the highest. When looking deeper into the data, I have two observations:

    a. The amount of datapoints is very small. Uncertainty bounds would be very useful in showing this data. Moreover, a discussion on how many datapoints are necessary for a reliable estimate in the first place would help the reader a lot.
    b. The data has been binned according to WS and WD, but not TI. This means that one bin can contain measurements with a very low turbulence intensity, and also measurements with a very high turbulence intensity. The underlying assumption of clubbing these data together in a single bin is that TI has no effect on the power production of the array of turbines (on page 20: "the effect of turbulence intensity is not expected to be as important"). I find this assumption to be unacceptable without a proper proof/validation/discussion.

    This assumption may likely explain the large variations in gains/losses seen in Figure 15. For example, if I compare the average of 3 hypothetical 'OFF' measurements with a very low turbulence intensity with 5 hypothetical 'ON' measurements with a very high turbulence intensity, I may see such large gains of tens of percent. However, this does not directly mean that the gains are due to axial induction control, but are also highly probable to be due to external factors (TI, in this example). Another explanation might be turbines turning on and off due to the slightest difference in wind speed, but I find this to be less likely. Actually, I believe Figure 18 supports my reasoning, since gains of similar magnitude are being predicted by LongSim. I strongly doubt whether LongSim (or any engineering wind farm model) in any setting would predict a gain of 70% solely due to axial induction control vs. greedy wind farm control. Rather, Figure 18 argues for the argument that the current data processing methodology does not accurately show the gain due to axial induction control in the field experiment.

    Using LongSim, you could check how much the relative power capture varies with wind

speed and with TI, and then choose to bin the data according to the one that varies the most (or both, if necessary). My guess would be that the relative gains do not significantly change with WS. If they do, and you have to bin the data according to both WS and TI, this may mean you are left with very few datapoints. Perhaps you have enough data for a handful of bins. Though, I would argue that it's better to have one or two reliable bins/gains than a large set of relatively unreliable gains.

Furthermore, I am missing a more detailed explanation about the results of the field experiment specifically. I understand that the entire process of synthesizing the controller is extremely important, but the focus seems to be a little bit off in this article. The results of the field test are not described until page 18 of the article. Then, the results are quite surprising and in my eyes deserve a detailed discussion. Finally, based on my reasoning above, I do not agree with the conclusion that axial induction control has been demonstrated/validated to increase the power production by several percent. The authors also state that statistically no quantification can be made, thereby seeming to contradict their own statement on the gains that can be achieved.

2. I believe the manuscript can be reduced in size, additionally improving readability. Namely, the authors tackle many topics and the paper therefore becomes very dense, yet I feel like it misses a clear direction at times.
    a. I believe this article should focus on the results of the axial induction control experiment. Validation of the LongSim model is very important and should, in my eyes, be a separate article.
    b. Sections such as 3.3 can be reduced significantly. The authors explain things in a very general sense, and then explain how they have done it themselves. I think the former is often not necessary (perhaps cite literature) or can be reduced to a minimum.
    c. Three different LUTs are compared in Section 4. The percentual increase in power production is 1.50%, 1.57% or 1.58%, respectively. These values fall within the uncertainty limits, no? To me, this seems like too much detail to show in the article. The authors could simply motivate the underlying ideas behind adding uncertainty/smearing (cite Rott, Quick, Simley), and state that smearing did not lead to losses in LongSim while alleviating the pitch actuation and increasing robustness.

3. An accurate reflection on the current literature on axial induction control is missing in this article. Namely, there are publications in the literature on axial induction control field experiments by ECN.TNO. There are also wind tunnel experiments by several researchers, among which Campagnolo et al. from TU Munich. The authors should explain what is new about this experiment, why it is necessary, place their findings in the appropriate context, and explain why their findings do or do not agree with the literature.

4. The article sometimes feels like a technical report more than it feels like a scientific article. Namely, the article contains descriptions of certain procedures and data which need not be mentioned for the understandability and reproducibility of the work (also, results cannot be reproduced anyhow since the data is not publicly available). For example,
    a. the paragraph on page 2 "The original intention […] Kern et al. (2019)." can be removed, in my eyes.
    b. the model acronyms in Figure 3 may be written in a more understandable way

c. Similarly, the manuscript contains statements about in which format the data was handed over from GE to DNV GL, and that it was updated periodically (page 17), which should be omitted in my eyes.

d. Also, the description on when certain turbines were curtailed is irrelevant for the results/conclusions in this manuscript. Figure 12 and the corresponding text can be removed; it suffices to state that curtailed data entries were removed from the dataset.

e. Generally, the text on page 18, section 5.1, can be reduced significantly. The authors can remove the following text snippets without any loss of generality or information: "namely the time stamp […] toggle state.", "as no setpoints … relevant records", "For the sake of … toggle flag changes", and "of the filter flag".

f. Rephrase statements such as "[…]possible to use measured stability as a lookup table input" to "[…] possible to measure stability".

5. I find it hard to read and understand several figures. Some figures miss an informative caption, labels on the axes, and the text in figures vary significantly in size throughout the document. For example,

a. Fig 3 has grey lines and a black box around it while other figures have black lines

b. Fig 4: the subplot titles in Fig 4: "A4-33/A4-38". The caption could read something along the lines of "Power production normalized by the power production of WTG 38", and the ylabel could read "Power ratio of WTG 37 [-]". Also the xlabels need not be repeated if the exact same axis/xlabel is used in the subplot directly underneath it. This makes the figure more compact. Generally, it would make sense to put Fig. 4 completely on one page, including caption, to avoid confusion.

c. Figure 6: "RotorAvDir" instead of "Rotor-averaged wind direction"

d. Figure 14: "results.mat", also missing units in labels

e. Figures 19, 20, 21: at this scale, it is not possible for the reader to draw any meaningful conclusions from them. Also, I believe these figures could be removed from the document to improve readability. Instead, I think tables with quantitative values would be much more interesting for validation.

f. Generally, legends would be appreciated in plots, though I understand that the captions also contain the information.

**Minor comments:**

- The title may be more informative: what kind of wind farm, what size of turbine array.
- The authors sometimes use vague language in the text. For example, "*convincing* validation", "*some* field tests", "the controller was toggled on and off at *regular* intervals", "the turbine is yawed *a little* out of the wind direction", "and *nearly* the lowest overall", "*almost* as good", "*higher*-frequency turbulence" (what frequencies?), "would be desirable to have *a lot* more datapoints", "the agreement is *very good*", "to reduce *some* individual turbine setpoints", "*excellent* agreement". I would be useful if the authors refrain from such vague statements, and rather use exact terms. For example, "the turbine is yawed, *typically by up to 30 degrees*, out of the wind direction*".
- Most of the references are technical documents of the CL-Windcon report. I suggest substituting these references as much as possible with scientific articles, though I understand this is difficult.

**Specific remarks:**

*Some of these comments may have already been addressed earlier, but just for completeness sake, I am putting down all the small things I have noticed here while reading the article:*

- Page 1: Abstract: *Horizon 20-20* or *Horizon 2020*?
- Page 1: Introduction: I am missing an inherent motivation of why this work is important. Perhaps the recent "Expert Elicitation on Wind Farm Control" paper by Wingerden et al. can help motivate this work. That paper shows a survey among experts which concludes that validation is currently the most important step before adoption by the industry.
- Page 1: A discussion on the effect of axial induction control on the pitch actuator duty cycle and the structural loads is missing. It would be good to address this, at least briefly.
- Page 2: Section 2: mention that it is an *onshore* wind farm.
- Page 2: Section 2: A wind rose and possibly a flow field from LongSim with wake interactions for WTG 31-38 could be insightful for the reader.
- Page 3: Figure 1: Why is there a blue box around this figure?
- Page 3: The citation "Knudsen et al." should be "van Wingerden et al." or "Doekemeijer et al.", considering TUDelft led this deliverable / report in CL-Windcon.
- Page 3, section 3.1: SCADA data was used for model comparison. Does this dataset contain turbine curtailment/derating? This would important to discuss in your article.
- Page 4: Perhaps add a citation for the bulk Richardson number
- Page 5: Perhaps add a citation for the Obukhov length
- Figure 2: Why does the x axis go until 25, rather than 24? Perhaps just hide 25.
- Figure 3: Missing units next to RMS, and remove the grey border around the figure
- Figure 4: The figure spans multiple pages, so when looking at page 6, it's not clear that the caption belonging to it is on page 8.
- Page 8: Explain the statement "However, for the purposes of the Sedini experiment this would not be possible to arrange".
- Page 9: Section 3.3: the first paragraph is written hypothetically. "in general", "would be", "can provide", "can usually", "can be", "could then". I would suggest the authors to keep a narrow focus on their own work, rather than explaining general methodologies/guidelines. A similar writing manner occurs in Section 3.4: "… in whatever measurements are actually used" and "wind conditions may not be the same".
- Page 10, section 3.4: when talking about including uncertainty in the optimization, perhaps cite Rott et al., Quick et al. and/or Simley et al. that have published on this topic.
- Page 10, "this has the advantage of faster optimisation, but also …". Perhaps it would be good to also discuss the disadvantages of smoothing the setpoints (and in this manner).
- Section 4: Generally, I would omit this section (in line with major comment 2). Sections such as 4.1 are general descriptions of LongSim and it might improve clarity if the authors would cite existing literature instead. If anything, important values can be collected in a table. In line with this, I think figures 5-13 contain too much information/would be better placed in a separate article.
- Figures 5 and 6 appear to be inconsistent. In Fig. 5, Turbine #38 is the title, while in Fig. 6, turbine #38 is part of the label. Also, the ylabel is missing and only units are given in Fig. 6. Generally, I suggest the authors to reconsider whether each figure is essential to the article and whether the style and size are consistent throughout.
- Figure 7 is missing an informative ylabel

- Figure 8: I can only see one line: the final smoothing one. Perhaps the authors can consider removing this figure and just stating this in their text.
- Figure 9 only has units on the ylabels, not the actual variable. The figure also appears before being mentioned in the text, which may be confusing to readers. Moreover, I suggest the authors to consider removing this figure.
- Figure 10: missing units on ylabel. Also, the purpose of this figure is unclear to me
- Figure 11 is missing a ylabel, and I suggest the authors to reconsider the need for this figure.
- Figure 12: may be removed, also not units or description on ylabels
- Figure 13 is missing a ylabel and units, xlabel should be **Time [hours]**, and may be removed
- Page 18: "at 1-minute resolution", this has been mentioned at least twice before in the article. I would suggest the authors to not repeat such information.
- Page 19: "it would clearly be desirable to have a lot more datapoints to give more confidence in the results", based on my reasoning in my major comment (1), I would expect that even with an infinitely large dataset, you will still not get rid of the large variations that you are seeing in the results. I think it would be essential to (at least) bin data according to turbulence intensity as well. (see major comment 1)
- Figure 14 contains no units on xlabel
- Page 20: the authors talk about a "mean" increase. Is this the mean of the mean of all bins? Is this the mean of all datapoints? Is this weighted according to the wind rose? Also, with variations of up to 70% and -40%, I have my doubts about taking a *mean*. The error margins would be very large. Please discuss this in your article.
- Figure 15 and 16 have differently sized text
- Page 22: "the predicted overall increase, 2.38% is very similar to the field test result of 2.42%". Though, from what I can see, the actual values between Fig 15 and 18 are quite significant. The fact that the mean values coincide does not serve as a validation by itself. It seems somewhat coincidence that the mean values are so close together.
- Figure 23: I was surprised the article went back to simulations here in Section 5.3.2 after having already discussed simulations until page 17, and then having discussed the field experiments.
- Figure 19, 20, 21: are these figures essential to the article? For me, I find it hard to derive any information from these figures, since they are so noisy and show data over such large timescales. Nonetheless, if the authors decide to keep these figures, please look at the ylabels and units. Also, in previous figures, units were given with square brackets around them. The font size also changes significantly between these figures.
- Page 25: generally, if a model validation is to be made in this article, I think quantification is essential. Perhaps first explain what is important: is it the absolute power by the turbines? Is it the relative power production compared to T1? Then how can you quantify the error/accuracy of the model? Do you want to validate steady-state or dynamic effects? I generally find that the authors make important steps in comparing their model, but I am missing a more informative discussion on what is necessary, how to quantify it, and the quantitative results.

---

## Referee Comment (RC2) · Stoyan Kanev (Referee) · 19 Aug 2020

Dear authors,

I found it very interesting to read your paper on the filed testing with induction control performed at Sedini wind farm within the CL-Windcon project. I was happy to see the positive results in terms of achieved power gain, even though the uncertainty in the estimates is quite high due to the low number of useful data points collected. Still, these results confirm earlier results from field testing with induction control (our own work, refer to paper of Daan van der Hoek or report by Koen Boorsma). These results still contradict with results from high-fidelity simulation and wind tunnel testing, and I believe it would be useful to make that point clear in the introduction. Please add some

relevant citations to put the paper in the right perspective.

Other minor comments:

- line 6: "20-20" -> "2020"

- page 1, line 28: Please, include reference to Bart Doekemeijer's paper about wake redirection control at Sedini wind farm.

- page 4, lines 76-79: you refer to Figures 3-4 here, while these have not yet been properly introduced in the text. I suggest you remove the references, or move these lines to a later point in the text.

- page 5, line 107: "Obukov length of -255" - please correct.

- page 5, lines 113-114: please provide a list with the compared models clarifying their main components in view of the model variations described in lines 91-100

- page 6, plots at bottom: please provide separate figure number for these plots.

- page 7, plots at bottom: please provide separate figure number for these plots.

- page 10, line 184: please provide a reference to earlier work on robust active wake control optimization including distributions

- page 17, line 316: "34" -> "33" (33 is curtailed according to Figure 12)

- page 17, Figure 12: please add units on y axis

- page 18, Figure 13: please add units on y axis

---

## Author Comment (AC2) · 28 Aug 2020

Thank you for your positive comments, and for your useful suggestions which we will use to improve the final version of the paper.

---

## Author Response (AR1)

**Review WES-2020-88**

Author responses to reviewer comments are in blue. Changes to the paper are in green.

**Reviewer 1 comments**

Dear authors,

I enjoyed reading your article on the field experiment at Sedini investigating axial induction control. I find this work to be extremely important in the further validation of wind farm control technologies. The article contains a lot of information and generally has a good story. Though, I have a number of major comments on the article.

10 I want to stress that these comments are my opinion, and with the sole intention of improving the scientific relevance of the manuscript. These comments are not meant to discourage the authors in their work, and I want to again emphasize that I am convinced of the article's importance. I hope this feedback finds you well.

**General comments:**

1. The core results of this article, based on the title and my interpretation of the article, would be represented

15 by Figures 15 and 16. These figures show the reported field experiment gains in power production using axial induction control compared to baseline operation. The authors show gains of up to 70% and losses of up to 40% using this algorithm. Based on what the literature has shown before on axial induction control, I find such values difficult to believe; We agree, and say so in the paper. I would expect the gains/losses to be in the order of 5% at the highest. When looking deeper into the data, I have two observations:

20 a. The amount of datapoints is very small. We definitely agree, and say so in the paper. Uncertainty bounds would be very useful in showing this data. Moreover, a discussion on how many datapoints are necessary for a reliable estimate in the first place would help the reader a lot. To do justice to such a discussion would require statistical theory, which would be a paper in itself.

b. The data has been binned according to WS and WD, but not TI. This means that one bin can contain

25 measurements with a very low turbulence intensity, and also measurements with a very high turbulence intensity. The underlying assumption of clubbing these data together in a single bin is that TI has no effect on the power production of the array of turbines We do not assume this – we simply present a result averaged across TIs because there is insufficient data to resolve the effect of TI. (on page 20: "the effect of turbulence intensity is not expected to be as important"). I find this assumption to be unacceptable without a proper

30 proof/validation/discussion. Not so much an assumption as a finding from the modelling: the higher the TI, the smaller the power gains, but the optimal setpoints do not change very much. By averaging over TI we will get a smaller increase than we would at the low TI values. In contrast, the effect of wind direction is fundamental - with setpoints varying greatly, as is wind speed, because the controller response to the setpoint varies significantly with wind speed. However, to avoid confusion, we have removed this phrase.

35 Your overall point about turbulence intensity is very important though, and we have included a significant amount of additional analysis in the final paper to investigate this and mitigate against bias, including extending Figure 14 (now 15) to show TI, and introducing a new Figure 19 and Table 3.

This assumption may likely explain the large variations in gains/losses seen in Figure 15. For example, if I

40 compare the average of 3 hypothetical 'OFF' measurements with a very low turbulence intensity with 5 hypothetical 'ON' measurements with a very high turbulence intensity, I may see such large gains of tens of percent. However, this does not directly mean that the gains are due to axial induction control, but are also highly probable to be due to external factors (TI, in this example). We agree, and (as in our reply to a previous short

comment) in the final version we explicitly mention TI combined with small numbers of points as a
45 contributory reason for the unrealistic numbers seen in some bins. Another explanation might be turbines
turning on and off due to the slightest difference in wind speed, but I find this to be less likely. Actually, I believe
Figure 18 supports my reasoning, since gains of similar magnitude are being predicted by LongSim. I strongly
doubt whether LongSim (or any engineering wind farm model) in any setting would predict a gain of 70% solely
due to axial induction control vs. greedy wind farm control. (We agree, see replies above.) Rather, Figure 18
50 argues for the argument that the current data processing methodology does not accurately show the gain due to
axial induction control in the field experiment. Absolutely, and we state that there isn't enough to data to get
statistically meaningful results in individual wind speed and direction bins, but it still helps in deriving as
much understanding as possible from the limited data we do have. The main lesson from Figure 18 is that
we get good agreement between the model and the measurements, which lends confidence in the use of
55 the model to design such controllers.

Using LongSim, you could check how much the relative power capture varies with wind speed and with TI, and
then choose to bin the data according to the one that varies the most (or both, if necessary). My guess would be
that the relative gains do not significantly change with WS. This is likely to be true over a limited wind speed
60 range only, because the action of the setpoints changes depending on windspeed. These details of the
turbine control algorithm are confidential, unfortunately. If they do, and you have to bin the data according to
both WS and TI, this may mean you are left with very few datapoints. Perhaps you have enough data for a
handful of bins. Though, I would argue that it's better to have one or two reliable bins/gains than a large set of
relatively unreliable gains. See replies above concerning the additional analysis which we have included in the final
65 version to cover this issue.

Furthermore, I am missing a more detailed explanation about the results of the field experiment specifically. I
understand that the entire process of synthesizing the controller is extremely important, but the focus seems to be
a little bit off in this article. The results of the field test are not described until page 18 of the article. Then, the
70 results are quite surprising and in my eyes deserve a detailed discussion. Finally, based on my reasoning above, I
do not agree with the conclusion that axial induction control has been demonstrated/validated to increase the
power production by several percent. The authors also state that statistically no quantification can be made,
thereby seeming to contradict their own statement on the gains that can be achieved. "Demonstrated" is arguably
fair, but the result is definitely not validated in a statistically meaningful way, as we say in the conclusions. In
75 any case, we have further elaborated the wording in the conclusions, also to cover the additional analysis
done as mentioned above.

2. I believe the manuscript can be reduced in size, additionally improving readability. Namely, the authors tackle
many topics and the paper therefore becomes very dense, yet I feel like it misses a clear direction at times.
80 a. I believe this article should focus on the results of the axial induction control experiment. Validation of the
LongSim model is very important and should, in my eyes, be a separate article.
b. Sections such as 3.3 can be reduced significantly. The authors explain things in a very general sense, and then
explain how they have done it themselves. I think the former is often not necessary (perhaps cite literature) or can
be reduced to a minimum. We have a different opinion concerning your points a and b. We wanted to
85 present a complete story covering the controller design and the field tests. We feel that validation of the
model used for the design is just as important as an outcome of the field tests.

c. Three different LUTs are compared in Section 4. The percentual increase in power production is 1.50%, 1.57% or 1.58%, respectively. These values fall within the uncertainty limits, no? To me, this seems like too much detail to show in the article. The authors could simply motivate the underlying ideas behind adding uncertainty/smearing (cite Rott, Quick, Simley), and state that smearing did not lead to losses in LongSim while alleviating the pitch actuation and increasing robustness. The fact the the values are so similar is actually a useful result, and we present these results to back this up. We have added references to the principle as you suggest.

3. An accurate reflection on the current literature on axial induction control is missing in this article. Namely, there are publications in the literature on axial induction control field experiments by ECN.TNO. There are also wind tunnel experiments by several researchers, among which Campagnolo et al. from TU Munich. The authors should explain what is new about this experiment, why it is necessary, place their findings in the appropriate context, and explain why their findings do or do not agree with the literature. Although we did not want to make the article even longer by including a full literature review, we have now included a brief discussion (with references) of previous results from field tests as well as wind tunnel and LES simulation experiments.

4. The article sometimes feels like a technical report more than it feels like a scientific article. Namely, the article contains descriptions of certain procedures and data which need not be mentioned for the understandability and reproducibility of the work (also, results cannot be reproduced anyhow since the data is not publicly available). For example,

a. the paragraph on page 2 "The original intention […] Kern et al. (2019)." can be removed, in my eyes. We actually feel that this background may be interesting to some readers. We have even expanded it slightly to refer to your recent paper on the wake steering tests at Sedini.

b. the model acronyms in Figure 3 may be written in a more understandable way We have added a table to identify the model variations represented by the acronyms.

c. Similarly, the manuscript contains statements about in which format the data was handed over from GE to DNV GL, and that it was updated periodically (page 17), which should be omitted in my eyes. This would not shorten the paper much, and we feel the information might help some readers to appreciate the sort of issues which typically arise during field test campaigns.

d. Also, the description on when certain turbines were curtailed is irrelevant for the results/conclusions in this manuscript. Figure 12 and the corresponding text can be removed; it suffices to state that curtailed data entries were removed from the dataset. This is not the case. As we state, the SCADA data did not contain curtailment flags to allow us to remove this data. Figure 12 illustrates a case where this could be important.

e. Generally, the text on page 18, section 5.1, can be reduced significantly. The authors can remove the following text snippets without any loss of generality or information: "namely the time stamp […] toggle state.", "as no setpoints … relevant records", "For the sake of … toggle flag changes", and "of the filter flag". Actually we disagree. The detail of how the data filtering is done is extremely important. The results can be significantly biased by filtering incorrectly, so we wanted to present this information openly.

f. Rephrase statements such as "[…]possible to use measured stability as a lookup table input" to "[…] possible to measure stability". We think this makes the sense less clear.

5. I find it hard to read and understand several figures. Some figures miss an informative caption, labels on the axes, and the text in figures vary significantly in size throughout the document. Thanks for your suggestions for
130    the figure captions and labels. We have dealt with these in the final version. For example,
a. Fig 3 has grey lines and a black box around it while other figures have black lines Done.
b. Fig 4: the subplot titles in Fig 4: "A4-33/A4-38". The caption could read something along the lines of "Power production normalized by the power production of WTG 38", and the ylabel could read "Power ratio of WTG 37 [-]". Also the xlabels need not be repeated if the exact same axis/xlabel is used in the subplot directly
135    underneath it. This makes the figure more compact. Generally, it would make sense to put Fig. 4 completely on one page, including caption, to avoid confusion. This figure has been updated
c. Figure 6: "RotorAvDir" instead of "Rotor-averaged wind direction" Done.
d. Figure 14: "results.mat", also missing units in labels Done.
e. Figures 19, 20, 21: at this scale, it is not possible for the reader to draw any meaningful conclusions from them.
140    Also, I believe these figures could be removed from the document to improve readability. Instead, I think tables with quantitative values would be much more interesting for validation. We feel that these figures give a useful indication of the ability of the model to represent reality, which is an important result for its application, for both algorithm design and field diagnostics. To provide quantitative values we would first have to decide on some useful values to present, and we feel that many readers would appreciate the graphical representation
145    more.
f. Generally, legends would be appreciated in plots, though I understand that the captions also contain the information. We have improved the detail of many of the figures.

**Minor comments:**
150    • The title may be more informative: what kind of wind farm, what size of turbine array. We do not want the title to become too long, and as there have been very few such field tests to date, we feel the shorter title encapsulates what is important.
• The authors sometimes use vague language in the text. For example, "*convincing* validation", "*some* field tests", "the controller was toggled on and off at *regular* intervals", "the turbine is yawed *a little* out of the wind
155    direction", "and *nearly* the lowest overall", "*almost* as good", "*higher*-frequency turbulence" (what frequencies?), "would be desirable to have *a lot* more datapoints", "the agreement is *very good*", "to reduce *some* individual turbine setpoints", "*excellent* agreement". I would be useful if the authors refrain from such vague statements, and rather use exact terms. For example, "the turbine is yawed, *typically by up to 30 degrees*, out of the wind direction". Thanks for the suggestion. Sometimes it is not possible to quantify things precisely, and vague
160    language is better than nothing, but we have been through and changed many of these in the final version.
• Most of the references are technical documents of the CL-Windcon report. I suggest substituting these references as much as possible with scientific articles, though I understand this is difficult. We definitely try to do this where it's possible.

165    **Specific remarks:** *Some of these comments may have already been addressed earlier, but just for completeness sake, I am putting down all the small things I have noticed here while reading the article:*
• Page 1: Abstract: *Horizon 20-20* or *Horizon 2020*? All changed to Horizon 2020.
• Page 1: Introduction: I am missing an inherent motivation of why this work is important. Perhaps the recent "Expert Elicitation on Wind Farm Control" paper by Wingerden et al. can help motivate this work. That paper
170    shows a survey among experts which concludes that validation is currently the most important step before

adoption by the industry. Thanks for the suggested reference. We have made significant changes to the introduction in line with your comment.

• Page 1: A discussion on the effect of axial induction control on the pitch actuator duty cycle and the structural loads is missing. It would be good to address this, at least briefly. We state that we ignore loads in this field

175 test because no measurements were available, but we have added a statement that loads and actuator duty are important considerations.

• Page 2: Section 2: mention that it is an *onshore* wind farm. We now state this early on.

• Page 2: Section 2: A wind rose and possibly a flow field from LongSim with wake interactions for WTG 31-38 could be insightful for the reader. Thanks for the suggestion. We have now included both of these.

180 • Page 3: Figure 1: Why is there a blue box around this figure? Thanks, we have removed the box.

• Page 3: The citation "Knudsen et al." should be "van Wingerden et al." or "Doekemeijer et al.", considering TUDelft led this deliverable / report in CL-Windcon. We fully sympathise with this comment, but Knudsen appears as the first-named of many authors of this report, presumably because the institutions are in alphabetical order.

185 • Page 3, section 3.1: SCADA data was used for model comparison. Does this dataset contain turbine curtailment/derating? This would important to discuss in your article. No, it doesn't, as we mention in the text associated with Figure 12 (now 13).

• Page 4: Perhaps add a citation for the bulk Richardson number Reference added

• Page 5: Perhaps add a citation for the Obukhov length Reference added

190 • Figure 2: Why does the x axis go until 25, rather than 24? Perhaps just hide 25.

• Figure 3: Missing units next to RMS, and remove the grey border around the figure Done.

• Figure 4: The figure spans multiple pages, so when looking at page 6, it's not clear that the caption belonging to it is on page 8. Sub-captions added for clarity.

195 • Page 8: Explain the statement "However, for the purposes of the Sedini experiment this would not be possible to arrange".

• Page 9: Section 3.3: the first paragraph is written hypothetically. "in general", "would be", "can provide", "can usually", "can be", "could then". I would suggest the authors to keep a narrow focus on their own work, rather than explaining general methodologies/guidelines. A similar writing manner occurs in Section 3.4: "… in

200 whatever measurements are actually used" and "wind conditions may not be the same". Changes have been made in the final version where appropriate.

• Page 10, section 3.4: when talking about including uncertainty in the optimization, perhaps cite Rott et al., Quick et al. and/or Simley et al. that have published on this topic. Thanks for your comment. Two references have been added.

205 • Page 10, "this has the advantage of faster optimisation, but also …". Perhaps it would be good to also discuss the disadvantages of smoothing the setpoints (and in this manner).

• Section 4: Generally, I would omit this section (in line with major comment 2). Sections such as 4.1 are general descriptions of LongSim and it might improve clarity if the authors would cite existing literature instead. If anything, important values can be collected in a table. In line with this, I think figures 5-13 contain too much

210 information/would be better placed in a separate article.

• Figures 5 and 6 appear to be inconsistent. In Fig. 5, Turbine #38 is the title, while in Fig. 6, turbine #38 is part of the label. Also, the ylabel is missing and only units are given in Fig. 6. Generally, I suggest the authors to

reconsider whether each figure is essential to the article and whether the style and size are consistent throughout.

215 • Figure 7 is missing an informative ylabel

• Figure 8: I can only see one line: the final smoothing one. Perhaps the authors can consider removing this figure and just stating this in their text. A note has been added to the figure's caption. We have updated many figures in line with the next few comments.

• Figure 9 only has units on the ylabels, not the actual variable. The figure also appears before being mentioned

220 in the text, which may be confusing to readers. Moreover, I suggest the authors to consider removing this figure.

• Figure 10: missing units on ylabel. Also, the purpose of this figure is unclear to me

• Figure 11 is missing a ylabel, and I suggest the authors to reconsider the need for this figure.

• Figure 12: may be removed, also not units or description on ylabels

225 • Figure 13 is missing a ylabel and units, xlabel should be **Time [hours]**, and may be removed

• Page 18: "at 1-minute resolution", this has been mentioned at least twice before in the article. I would suggest the authors to not repeat such information.

• Page 19: "it would clearly be desirable to have a lot more datapoints to give more confidence in the results", based on my reasoning in my major comment (1), I would expect that even with an infinitely large dataset, you

230 will still not get rid of the large variations that you are seeing in the results. I think it would be essential to (at least) bin data according to turbulence intensity as well. (see major comment 1) Good point, and we have extended the analysis as already explained above.

• Figure 14 contains no units on xlabel

• Page 20: the authors talk about a "mean" increase. Is this the mean of the mean of all bins? Is this the mean of

235 all datapoints? Is this weighted according to the wind rose? Also, with variations of up to 70% and -40%, I have my doubts about taking a *mean*. The error margins would be very large. Please discuss this in your article. This is the unweighted mean of all the data points, presented 'for what it's worth', given the paucity of data, with a statement that no special significance should be attached to the precise numbers.

• Figure 15 and 16 have differently sized text

240 • Page 22: "the predicted overall increase, 2.38% is very similar to the field test result of 2.42%". Though, from what I can see, the actual values between Fig 15 and 18 are quite significant. The fact that the mean values coincide does not serve as a validation by itself. It seems somewhat coincidence that the mean values are so close together. Validation is not yes/no. We believe that these results, together with everything else, contribute to an increase in confidence in the model.

245 • Figure 23: I was surprised the article went back to simulations here in Section 5.3.2 after having already discussed simulations until page 17, and then having discussed the field experiments. The point of 5.3.2 is the use of field data for validating the time-domain aspects of the modelling.

• Figure 19, 20, 21: are these figures essential to the article? For me, I find it hard to derive any information from these figures, since they are so noisy and show data over such large timescales. Nonetheless, if the

250 authors decide to keep these figures, please look at the ylabels and units. Also, in previous figures, units were given with square brackets around them. The font size also changes significantly between these figures.

• Page 25: generally, if a model validation is to be made in this article, I think quantification is essential. Perhaps first explain what is important: is it the absolute power by the turbines? Is it the relative power production compared to T1? Then how can you quantify the error/accuracy of the model? Do you want to validate steady-

255 state or dynamic effects? I generally find that the authors make important steps in comparing their model, but I

am missing a more informative discussion on what is necessary, how to quantify it, and the quantitative results. You are right of course, but this could fill a whole paper.

**Reviewer 2 comments**

260

Dear authors,
I found it very interesting to read your paper on the filed testing with induction control performed at Sedini wind farm within the CL-Windcon project. I was happy to see the positive results in terms of achieved power gain, even though the uncertainty in the

265 estimates is quite high due to the low number of useful data points collected. Still, these results confirm earlier results from field testing with induction control (our own work, refer to paper of Daan van der Hoek or report by Koen Boorsma). These results still contradict with results from high-fidelity simulation and wind tunnel testing, and I believe it would be useful to make that point clear in the introduction. Please add some

270 relevant citations to put the paper in the right perspective. Thank you for the comment. We have added references and comments relating to the paper of Daan van der Hoek and also previous wind tunnel tests and LES simulations.
Other minor comments:
• line 6: "20-20" -> "2020" Changed.
• page 1, line 28: Please, include reference to Bart Doekemeijer's paper about

275 wake redirection control at Sedini wind farm. A reference to this paper has been added (now on page 2).
• page 4, lines 76-79: you refer to Figures 3-4 here, while these have not yet been properly introduced in the text. I suggest you remove the references, or move these lines to a later point in the text. We have reconfigured this. Hopefully it is clearer now.
• page 5, line 107: "Obukov length of -255" - please correct. Done.

280 • page 5, lines 113-114: please provide a list with the compared models clarifying their main components in view of the model variations described in lines 91-100 Done, thanks for the suggestion.
• page 6, plots at bottom: please provide separate figure number for these plots.
• page 7, plots at bottom: please provide separate figure number for these plots. Captions for these partial figures have been added for clarity.

285 • page 10, line 184: please provide a reference to earlier work on robust active wake control optimization including distributions This has been done (see response to Reviewer 1 on this).
• page 17, line 316: "34" -> "33" (33 is curtailed according to Figure 12) This has been corrected.
• page 17, Figure 12: please add units on y axis Done.
• page 18, Figure 13: please add units on y axis Done.

290

A marked-up copy of the paper follows:

[revised manuscript text omitted]

---

## Referee Report (RR1)

Dear authors,

First of all, thank you for your responses to my comments. You have taken my points seriously and answered my questions with care. The figures have significantly improved, the document is easier to read, and notes and explanations have been added contextually. Also, not all comments led to a change in the manuscript, and I fully understand that.

Though, one important concern of me remains with the current publication. The conclusion of this article invokes the idea that axial induction control provided an estimated gain in power production of 1.7% to 2.4%. I am not saying that I disagree with these numbers, but at the same time I am not yet fully convinced after reading the paper. As mentioned before and as you have discussed in your manuscript, there may be other factors at play. I believe you can investigate this more, which would bring more confidence to these numbers.  When I look at figure 15, I see that for basically every bin, the mean turbulence intensity for ON is higher than OFF. A higher turbulence intensity may give a higher power production in these situations. This argument is supported by looking at bins with directions 185 to 205 degrees, where axial induction control is not really applied and yet gains are large. Similarly, when looking at the left subplots of Figure 15, the largest gain is at the bin with WS = 10.5 m/s, which also has the largest difference in TI between ON/OFF. At other bins with almost equal TI between ON/OFF, we also see barely any gain in power production.

Now, when we look at the right subplots of Figure 15, things get better and I think this does indicate a gain due to axial induction control, specifically for bins 235 deg and 255 deg, which have almost no difference in TI. I believe it is fair to say that these bins indicate a gain, but I think it is not fair to say that *every* bin indicates a gain due to axial induction control. I think one must investigate the effect of the turbulence intensity (among others) first, before being able to conclude this. Similarly, if you use all data to calculate mean values like on page 21, then you will be comparing ON data with an effectively higher mean TI with OFF data with an effectively lower mean TI. There may indeed be a 2.42% gain between ON and OFF, but perhaps this is 2% due to TI and only 0.42% due to axial induction control. I would suggest the authors to be very careful making such statements. I anticipate this to be a high-impact paper and these numbers are likely to be used as reference values for the potential of axial induction control.

Now, the additional figures, especially Figure 18, adds significant value to the manuscript. From this figure, I conclude that the mean TI between ON and OFF is not very large. This is a very interesting observation and supports the conclusion that the authors already make.

To strengthen the conclusions from the authors, I would suggest diving deeper into Figures 16 and 20. LongSim was used to (re)simulate the measurement points and also reported significant gains. Using LongSim, you could figure out where these large gains come from. If in the "ON" dataset you simulate them with the baseline (OFF) controller, do you still see such large gains? If so, then the gains are not due to axial induction control, but due to other effects. Similarly, what if you simulate the datapoints in LongSim with a fixed value for TI – that could provide insight into the effect of the turbulence intensity on the power gain. By doing these manual simulations, you will gain much insight into where the gains are really coming from, and perhaps why the values are so high. Since such large gains are also seen in LongSim, it must be more than statistical uncertainty.

A second idea to give more insight into the effect of TI is by redistributing the bins in Figure 15 so make sure their average TI is identical. For example, one could make duplicates of low-TI "ON" measurements or make duplicates of high-TI "OFF" measurements to bring the average TI to the same value. Now, with the datasets already being so sparse, one may introduce additional bias, so perhaps the former suggestion is better.

Figuring this out will also prevent vague statement such as on page 20 "it is possible that [...] might account for [..] some [...] power increase."

Smaller comments:
- Abstract: "show a positive increase in energy production resulting from induction control" is a strong statement. If anything, perhaps rephrase it to something like "the experimental data suggests that induction control leads to both gains and losses in power production, with the gains outnumbering the losses."
- Figure 14: for consistency, it would make sense to put the plot title on the ylabel instead. Same goes for Figure 21. Also, for Figure 21, perhaps change the xlabel to "Time [Hours]"
- I am not sure if Table 3 adds significant value
- Figure 23: the authors state that the agreement is very good. Though, I believe at this scale it's hard to draw this conclusion. The power production may be off by 30% at any point in time, especially at such low absolute power values.

---

## Referee Report (RR2)

The article is interesting, well written, and deals with the field experimentation of a cooperative wind farm control strategy that aims at increasing the overall power of a wind farm through an appropriate reduction of the power produced by the upstream machines, whose effect is to reduce their axial induction and thus to mitigate the speed deficit in their wakes. Precedent experiments conducted in wind tunnel and simulated environment (properly cited by the authors) have shown that this method seems to be not very effective. As well as the previous experiments conducted in real life have shown extremely small gains in the order of the measurement uncertainty, leading to inconclusive results.

This paper has the merit to describe in very detailed way the experimental setup and the methodology used for the synthesis and the implementation of the controller, as well as the simulation model used to the purpose. For this reason, I think that it is worthy to be published in the journal.

As stated by the authors, the data obtained through the experimentation are not sufficient to derive statistically robust results. Moreover, the used simulation model shows a partial agreement with the experimental data, especially regarding the predictions for the machines located more downstream in the array. The differences between the model-predicted and the experimental overall-cluster power (not shown in the paper, even if they should be) are probably superior to the power gains measured experimentally, a fact that suggests that the model cannot be used to validate the obtained results.

I believe, therefore, that the results obtained do not allow to conclude on the effectiveness or not of the tested method in terms of boosting the wind farm power output. This aspect should be emphasized in the conclusions, which should also include what should be, according to the authors, the actions required in order to reach a satisfactory conclusion on the effectiveness or not of the proposed method.

I also report in the following some other suggestions for improving the manuscript.

- Page 1, line 21-22. The control concepts the sentence refers to are not introduced before.
- Section 3.2 what exactly the control set-point is? Is it the power reduction, expressed in percent of the available one? I think it is important that the authors clarified this aspect. Moreover, I think it is important to show here some resulting LUT, and quickly comment how close the computed optimal set-points are with respect to those adopted by other authors whose findings have been cited in the introduction.
- Page 18-Line 359: to allow the reader understanding why what is shown in Fig 14 is a "small set-point", it must be clarified before what the set-point is.
- Page 26, Line 669-470. I would not claim that 20 degrees of difference in the wind direction is a slight difference. The wake-to-turbines interaction with 225 degree wind direction is totally different from the one with 245 wind direction.
- Figure 20. It would be very beneficial to put aside of Figure 20 a similar figure that shows the delta between measured and simulated "power ratio ON/OFF". There is the space for it, and it would allow the reader to quickly get how good the predictions of power gains are with respect to the measured data.

- Page 28- Figure22, WT13 data. I would not claim that the agreement is that good as it was for WT 38-37 and, partially, also for WT36. In many instants, indeed, the predicted power is more than double the SCADA data. This probably means that the simulation model is overestimating the wake recovery as we look further downstream in the WTs array.
- Fig. 22: it would be very interesting to show here also the comparison between the measured and simulated overall cluster power. The control set-point are indeed derived with the goal of maximizing the overall cluster power, and LongSim is used as simulation model. it would be therefore very interesting to check how good the model is in capturing this quantity.
- Fig. 22 and 23: I personally don't think that visually comparing time series is the best way to judge the agreement between numerical and experimental data. I would have instead plotted the numerical data w.r.t. the experimental data, also including the correlation factor and the related RMS.
- For what shown in the paper, I would not be confident claiming that LongSim provided an excellent agreement with the experimental data for all WTs. Fig 22, for example, clearly shows that WT13 power is quite often overestimated.

---

## Author Response (AR2)

**Response to referee report 2 (Responses shown with yellow highlight)**

Dear authors,
First of all, thank you for your responses to my comments. You have taken my points seriously and answered my questions with care. The figures have significantly improved, the document is easier to read, and notes and explanations have been added contextually. Also, not all comments led to a change in the manuscript, and I fully understand that.

Though, one important concern of me remains with the current publication. The conclusion of this article invokes the idea that axial induction control provided an estimated gain in power production of 1.7% to 2.4%. I am not saying that I disagree with these numbers, but at the same time I am not yet fully convinced after reading the paper. As mentioned before and as you have discussed in your manuscript, there may be other factors at play. I believe you can investigate this more, which would bring more confidence to these numbers. When I look at figure 15, I see that for basically every bin, the mean turbulence intensity for ON is higher than OFF. A higher turbulence intensity may give a higher power production in these situations. This argument is supported by looking at bins with directions 185 to 205 degrees, where axial induction control is not really applied and yet gains are large. Similarly, when looking at the left subplots of Figure 15, the largest gain is at the bin with WS = 10.5 m/s, which also has the largest difference in TI between ON/OFF. At other bins with almost equal TI between ON/OFF, we also see barely any gain in power production.

Now, when we look at the right subplots of Figure 15, things get better and I think this does indicate a gain due to axial induction control, specifically for bins 235 deg and 255 deg, which have almost no difference in TI. I believe it is fair to say that these bins indicate a gain, but I think it is not fair to say that *every* bin indicates a gain due to axial induction control. I think one must investigate the effect of the turbulence intensity (among others) first, before being able to conclude this. Similarly, if you use all data to calculate mean values like on page 21, then you will be comparing ON data with an effectively higher mean TI with OFF data with an effectively lower mean TI. There may indeed be a 2.42% gain between ON and OFF, but perhaps this is 2% due to TI and only 0.42% due to axial induction control. I would suggest the authors to be very careful making such statements. I anticipate this to be a high-impact paper and these numbers are likely to be used as reference values for the potential of axial induction control. An additional comment has been added to emphasise this. I think it's already stated clearly enough for the right-hand plot. A similar comment has been added to the conclusions.

Now, the additional figures, especially Figure 18, adds significant value to the manuscript. From this figure, I conclude that the mean TI between ON and OFF is not very large. This is a very interesting observation and supports the conclusion that the authors already make.

To strengthen the conclusions from the authors, I would suggest diving deeper into Figures 16 and 20. LongSim was used to (re)simulate the measurement points and also reported significant gains. Using LongSim, you could figure out where these large gains come from. If in the "ON" dataset you simulate them with the baseline (OFF) controller, do you still see such large gains? If so, then the gains are not due to axial induction control, but due to other effects. Similarly, what if you simulate the datapoints in LongSim with a fixed value for TI – that could provide insight into the effect of the turbulence intensity on the power gain. By doing these manual simulations, you will gain much insight into where the gains are really coming from, and perhaps why the values are so high. Since such large gains are also seen in LongSim, it must be more than statistical uncertainty. Actually I did some investigations along these lines before submitting the last version, but it was a bit inconclusive, and not useful enough to be worth making the paper even longer.

A second idea to give more insight into the effect of TI is by redistributing the bins in Figure 15 so make sure their average TI is identical. For example, one could make duplicates of low-TI "ON" measurements or make duplicates of high-TI "OFF" measurements to bring the average TI to the same value. Now, with the datasets already being so sparse, one may introduce additional bias, so perhaps the former suggestion is better. Figuring this out will also prevent vague statement such as on page 20 "it is possible that […] might account for […] some […] power increase."

Smaller comments:

•        Abstract: "show a positive increase in energy production resulting from induction control" is a strong statement. If anything, perhaps rephrase it to something like "the experimental data suggests that induction control leads to both gains and losses in power production, with the gains outnumbering the losses." Some additional words have been added at line 15 about there not being enough data

•        Figure 14: for consistency, it would make sense to put the plot title on the ylabel instead. y-axis label has been addedSame goes for Figure 21. Also, for Figure 21, perhaps change the xlabel to "Time [Hours]" Done

•        I am not sure if Table 3 adds significant value I am surprised at the comment, as the table was added to provide more transparency, as a way to deal with previous comments from this reviewer, to help readers draw their own conclusions.

•        Figure 23: the authors state that the agreement is very good. Though, I believe at this scale it's hard to draw this conclusion. The power production may be off by 30% at any point in time, especially at such low absolute power values. A note has been added around line 500 about expected discrepancies at higher frequencies, etc. which we believe answers this point.

**Response to referee report 4 (Responses shown with yellow highlight)**

 The article is interesting, well written, and deals with the field experimentation of a cooperative wind farm control strategy that aims at increasing the overall power of a wind farm through an appropriate reduction of the power produced by the upstream machines, whose effect is to reduce their axial induction and thus to mitigate the speed deficit in their wakes. Precedent experiments conducted in wind tunnel and simulated environment (properly cited by the authors) have shown that this method seems to be not very effective. As well as the previous experiments conducted in real life have shown extremely small gains in the order of the measurement uncertainty, leading to inconclusive results.

This paper has the merit to describe in very detailed way the experimental setup and the methodology used for the synthesis and the implementation of the controller, as well as the simulation model used to the purpose. For this reason, I think that it is worthy to be published in the journal.

As stated by the authors, the data obtained through the experimentation are not sufficient to derive statistically robust results. Moreover, the used simulation model shows a partial agreement with the experimental data, especially regarding the predictions for the machines located more downstream in the array. The differences between the model-predicted and the experimental overall-cluster power (not shown in the paper, even if they should be) are probably superior to the power gains measured experimentally, a fact that suggests that the model cannot be used to validate the obtained results.

I believe, therefore, that the results obtained do not allow to conclude on the effectiveness or not of the tested method in terms of boosting the wind farm power output. This aspect should be emphasized in the conclusions, A further comment has been added to the conclusions. which should also include what should

be, according to the authors, the actions required in order to reach a satisfactory conclusion on the effectiveness or not of the proposed method. A further sentence added to the conclusions for this. I also report in the following some other suggestions for improving the manuscript.

・ Page 1, line 21-22. The control concepts the sentence refers to are not introduced before. This has been corrected

・ Section 3.2 what exactly the control set-point is? Is it the power reduction, expressed in percent of the available one? This is described, in as much detail as permitted by the manufacturer, on page 13 (line 267) I think it is important that the authors clarified this aspect. Moreover, I think it is important to show here some resulting LUT, and quickly comment how close the computed optimal set-points are with respect to those adopted by other authors whose findings have been cited in the introduction. This is not possible as the setpoints are completely dependent on layout and spacing, so there are no comparable examples in the literature.

・ Page 18-Line 359: to allow the reader understanding why what is shown in Fig 14 is a "small set-point", it must be clarified before what the set-point is. The definition of the setpoint is dealt with as far as possible as explained in reply to the second bullet point. The reference to small values of the setpoint has been removed  as it does not add anything significant.

・ Page 26, Line 669-470. I would not claim that 20 degrees of difference in the wind direction is a slight difference. The wake-to-turbines interaction with 225 degree wind direction is totally different from the one with 245 wind direction. the word 'slightly' has been deleted

・ Figure 20. It would be very beneficial to put aside of Figure 20 a similar figure that shows the delta between measured and simulated "power ratio ON/OFF". There is the space for it, and it would allow the reader to quickly get how good the predictions of power gains are with respect to the measured data. I don't think this would add much. The reader can get the full picture by visually comparing figures 20 and 16. A plot of the differences is more noisy and not so easy to interpret.

・ Page 28- Figure22, WT13 data. I would not claim that the agreement is that good as it was for WT 38-37 and, partially, also for WT36. In many instants, indeed, the predicted power is more than double the SCADA data. This probably means that the simulation model is overestimating the wake recovery as we look further downstream in the WTs array. An additional comment about this has been added around line 500.

・ Fig. 22: it would be very interesting to show here also the comparison between the measured and simulated overall cluster power. The control set-point are indeed derived with the goal of maximizing the overall cluster power, and LongSim is used as simulation model. it would be therefore very interesting to check how good the model is in capturing this quantity. This extra plot has been included.

・ Fig. 22 and 23: I personally don't think that visually comparing time series is the best way to judge the agreement between numerical and experimental data. I would have instead plotted the numerical data w.r.t. the experimental data, also including the correlation factor and the related RMS. Because the unmeasured high frequencies are synthesised, small mismatches in time would lead to large scatter in such plots which does not reflect on model accuracy. The time series plots allow one to see such effects more clearly.

・ For what shown in the paper, I would not be confident claiming that LongSim provided an excellent agreement with the experimental data for all WTs. Fig 22, for example, clearly shows that WT13 power is quite often overestimated. I have already commented on this.